# Do Input Gradients Highlight Discriminative Features?

**Harshay Shah**[*]
Microsoft Research India
harshay@google.com

**Prateek Jain**[*]
Microsoft Research India
prajain@google.com

**Praneeth Netrapalli**[*]
Microsoft Research India
pnetrapalli@google.com

## Abstract

Post-hoc gradient-based interpretability methods [1, 2] that provide instance-specific explanations of model predictions are often based on assumption (A): *magnitude of input gradients—gradients of logits with respect to input—noisily highlight discriminative task-relevant features*. In this work, we test the validity of assumption (A) using a three-pronged approach:

1. We develop an evaluation framework, `DiffROAR`, to test assumption (A) on four image classification benchmarks. Our results suggest that (i) input gradients of standard models (i.e., trained on original data) may grossly violate (A), whereas (ii) input gradients of adversarially robust models satisfy (A) reasonably well.

2. We then introduce `BlockMNIST`, an `MNIST`-based semi-real dataset, that *by design* encodes a priori knowledge of discriminative features. Our analysis on `BlockMNIST` leverages this information to validate as well as characterize differences between input gradient attributions of standard and robust models.

3. Finally, we theoretically prove that our empirical findings hold on a simplified version of the `BlockMNIST` dataset. Specifically, we prove that input gradients of standard one-hidden-layer MLPs trained on this dataset do not highlight instance-specific "signal" coordinates, thus grossly violating (A).

Our findings motivate the need to formalize and test common assumptions in interpretability in a falsifiable manner [3]. We believe that the `DiffROAR` framework and `BlockMNIST` datasets serve as sanity checks to audit interpretability methods; code and data available at `https://github.com/harshays/inputgradients`.

## 1 Introduction

Interpretability methods that provide instance-specific explanations of model predictions are often used to identify biased predictions [4], debug trained models [5], and aid decision-making in high-stakes domains such as medical diagnosis [6, 7]. A common approach for providing instance-specific explanations is *feature attribution*. Feature attribution methods rank or score input coordinates, or features, in the order of their *purported* importance in model prediction; coordinates achieving the top-most rank or score are considered most important for prediction, whereas those with the bottom-most rank or score are considered least important.

**Input gradient attributions**. Ranking input coordinates based on the *magnitude of input gradients* is a fundamental feature attribution technique [8, 1] that undergirds well-known methods such as SmoothGrad [2] and Integrated Gradients [9]. Given instance $x$ and a trained model $\theta$ with prediction $\hat{y}$ on $x$, the input gradient attribution scheme (i) computes the input gradient $\nabla_x \text{Logit}_\theta(x, \hat{y})$ of the logit [2] of the predicted label $\hat{y}$ and (ii) ranks the input coordinates in *decreasing* order of their input gradient magnitude. Below we explicitly characterize the underlying intuitive assumption behind input gradient attribution methods:

---

[*]Part of the work completed after joining Google Research India

[2]In Appendix C, we show that our results also hold for input gradients taken w.r.t. the *loss*

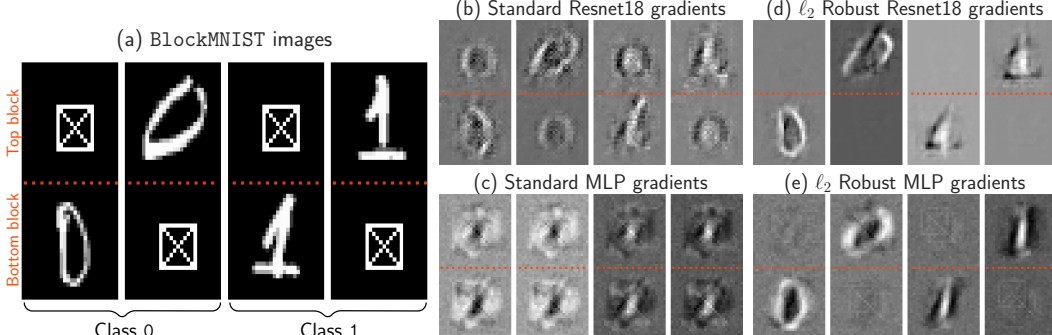

Figure 1: Experiments on `BlockMNIST` dataset. (a) Four representative images from class 0 & class 1 in `BlockMNIST` dataset; every image consists of a *signal* and *null* block that are randomly placed as the *top* or *bottom* block. The *signal* block, containing the `MNIST` digit, determines the image class. The *null* block, containing the square patch, does not encode any information of the image class. For these four images, subplots (b-e) show the input gradients of standard Resnet18, standard MLP, $\ell_2$ robust Resnet18 ($\epsilon$=2) and $\ell_2$ robust MLP ($\epsilon$=4) respectively. The plots clearly show that input gradients of standard `BlockMNIST` models highlight the signal block *and the non-discriminative null block*, thereby violating (A). In contrast, input gradients of adversarially robust models exclusively highlight the signal block, suppress the null block, and satisfy (A). Please see Section 5 for details.

> **Assumption** (A): *Coordinates with larger input gradient magnitude are more relevant for model prediction compared to coordinates with smaller input gradient magnitude.*

**Sanity-checking attribution methods**. Several attribution methods [10] are based on input gradients and explicitly or implicitly assume an appropriately modified version of (A). For example, Integrated Gradients [9] aggregate input gradients of linearly interpolated points, SmoothGrad [2] averages input gradients of points perturbed using gaussian noise, and Guided Backprop [11] modifies input gradients by zeroing out negative values at every layer during backpropagation. Surprisingly, unlike vanilla input gradients, popular methods that output attributions with better visual quality fail simple sanity checks that are indeed expected out of any valid attribution method [12, 13]. On the other hand, while vanilla input gradients pass simple sanity checks, Hooker et al. [14] suggest that they produce estimates of feature importance that are no better than a random designation of feature importance.

**Do input gradients satisfy assumption (A)?** Since (A) is necessary for input gradients attributions to accurately reflect model behavior, we introduce an evaluation framework, `DiffROAR`, to analyze whether input gradient attributions satisfy assumption (A) on real-world datasets. While `DiffROAR` adopts the remove-and-retrain (`ROAR`) methodology [14], `DiffROAR` is more appropriate for testing the validity of assumption (A) because it directly compares top-ranked features against bottom-ranked features. We apply `DiffROAR` to evaluate input gradient attributions of MLPs & CNNs trained on multiple image classification datasets. Consistent with the message in Hooker et al. [14], our experiments indicate that input gradients of standard models (i.e., trained on original data) can grossly violate (A) (see Section 4). Furthermore, we also observe that unlike standard models, adversarially trained models [15] that are robust to $\ell_2$ and $\ell_\infty$ perturbations satisfy (A) in a consistent manner.

**Probing input gradient attributions using `BlockMNIST`.** Our empirical findings mentioned above strongly suggest that standard models grossly violate (A). However, without knowledge of ground-truth discriminative features learned by models trained on real data, *conclusively* testing (A) remains elusive. In fact, this is a key shortcoming of the remove-and-retrain (`ROAR`) framework. So, to further verify and better understand our empirical findings, we introduce an `MNIST`-based semi-real dataset, `BlockMNIST`, that *by design* encodes a priori knowledge of ground-truth discriminative features. `BlockMNIST` is based on the principle that for different inputs, discriminative and non-discriminative features may occur in different parts of the input. For example, in an object classification task, the object of interest can occur in different parts of the image (e.g., top-left, center, bottom-right etc.) for different images. As shown in Figure 1(a), `BlockMNIST` images consist of a *signal* block and a *null* block that are randomly placed at the top or bottom. The *signal* block contains the `MNIST` digit that determines the class of the image, whereas the *null* block, contains a square patch with two diagonals that has no information about the label. This a priori knowledge of ground-truth discriminative

features in `BlockMNIST` data allows us to (i) validate our empirical findings vis-a-vis input gradients of standard and robust models (see fig. 1) and (ii) identify *feature leakage* as a reason that potentially explains why input gradients violate (A) in practice. Here, feature leakage refers to the phenomenon wherein given an instance, its input gradients highlight the location of discriminative features in the given instance *as well as* in other instances that are present in the dataset. For example, consider the first `BlockMNIST` image in fig. 1(a), in which the signal is placed in the bottom block. For this image, as shown in fig. 1(b,c), input gradients of standard models incorrectly highlight the top block *because* there are *other* instances in the `BlockMNIST` dataset which have signal in the top block.

**Rigorously demonstrating feature leakage**. In order to concretely verify as well as understand feature leakage more thoroughly, we design a simplified version of `BlockMNIST` that is amenable to theoretical analysis. On this dataset, we first rigorously demonstrate that input gradients of standard one-hidden-layer MLPs exhibit feature leakage in the infinite-width limit and then discuss how feature leakage results in input gradient attributions that clearly violate assumption (A).

**Paper organization**: Section 2 discusses related work and section 3 presents our evaluation framework, `DiffROAR`, to test assumption (A). Section 4 employs `DiffROAR` to evaluate input gradient attributions on four image classification datasets. Section 5 analyzes `BlockMNIST` data to differentially characterize input gradients of standard and robust models using feature leakage. Section 6 provides theoretical results on a simplified version on `BlockMNIST` that shed light on how feature leakage results in input gradients that violate assumption (A). Our code, along with the proposed datasets, is publicly available at https://github.com/harshays/inputgradients.

## 2  Related work

Due to space constraints, we only discuss directly related work and defer the rest to Appendix A.

**Sanity checks for explanations**. Several explanation methods that provide feature attributions are often primarily evaluated using inherently subjective visual assessments [1, 2]. Unsurprisingly, recent "sanity checks" show that sole reliance on visual assessment is misleading, as attributions can lack fidelity and inaccurately reflect model behavior. Adebayo et al. [12] and Kindermans et al. [13] show that unlike input gradients [8], other popular methods—guided backprop [16], gradient $\odot$ input [17], integrated gradients [9]—output explanations which lack fidelity on image data, as they remain invariant to model and label randomization. Similarly, Yang and Kim [18] use custom image datasets to show that several explanation methods are more likely to produce false positive explanations than vanilla input gradients. Moreover, several explanation methods based on modified backpropagation do not pass basic sanity checks [19, 20, 21]. To summarize, well-known gradient-based attribution methods that seek to mitigate gradient saturation [9, 22], discontinuity [23], and visual noise [16] surprisingly fare worse than vanilla input gradients on multiple sanity checks.

**Evaluating explanation fidelity**. The black-box nature of neural networks necessitates frameworks that evaluate the fidelity or "correctness" of post-hoc explanations *without* knowledge of ground-truth features learned by trained models. Modification-based evaluation frameworks [24, 25, 26] gauge explanation fidelity by measuring the change in model performance after masking input coordinates that a given explanation method considers most (or least) important. However, due to distribution shifts induced by input modifications, one cannot *conclusively* attribute changes in model performance to the fidelity of instance-specific explanations [27]. The remove-and-retrain (`ROAR`) framework [14] accounts for distribution shifts by evaluating the performance of models *retrained* on train data masked using post-hoc explanations. Surprisingly, contrary to findings obtained via sanity checks, experiments with the `ROAR` framework show that multiple attribution methods, *including* vanilla input gradients, are no better than model-independent *random* attributions that lack explanatory power [14]. Therefore, motivated by the central role of vanilla input gradients in attribution methods, we augment the `ROAR` framework to understand when and why input gradients violate assumption (A).

**Effect of adversarial robustness**. Adversarial training [15] not only leads to robustness to $\ell_p$ adversarial attacks [28], but also leads to perceptually-aligned feature representations [29], and improved visual quality of input gradients [30]. Recent works hypothesize that adversarial training improves the visual quality of input gradients by suppressing irrelevant features [31] and promoting sparsity and stability [32] in explanations. Kim et al. [33] use the `ROAR` framework to conjecture that adversarial training "tilts" input gradients to better align with the data manifold. In this work, we use experiments on real-world data and theory on data with features known *a priori* in order to differentially characterize input gradients of standard and robust models vis-a-vis assumption (A).

## 3  `DiffROAR` **evaluation framework**

In this section, we introduce our evaluation framework, `DiffROAR`, to probe the extent to which instance-specific explanations, or feature attributions, highlight discriminative features in practice. Specifically, our framework, `DiffROAR`, builds upon the remove-and-retrain (`ROAR`) methodology [14] to test whether feature attribution methods satisfy assumption (A) on real-world datasets.

**Setting**. We consider the standard classification setting; Each data point $(x^{(i)}, y^{(i)})$, where instance $x^{(i)} \in \mathbb{R}^d$ and label $y^{(i)} \in \mathcal{Y}$ for some label set $\mathcal{Y}$, is drawn independently from a distribution $\mathcal{D}$ on $\mathbb{R}^d \times \mathcal{Y}$. Given dataset $\{(x^{(i)}, y^{(i)})\}$ where $i \in [n] := \{1, \cdots, n\}$, $x_j^{(i)}$ denotes the $j^{\text{th}}$ coordinate of $x^{(i)}$. Note that we also refer to the $d$ coordinates of instance $x^{(i)}$ as *features* interchangeably.

**Attribution schemes**. A *feature attribution* scheme $\mathcal{A} : \mathbb{R}^d \to \{\sigma : \sigma \text{ is a permutation of } [d]\}$ maps a $d$-dimensional instance $x$ to a permutation, or ordering, $\mathcal{A}(x) : [d] \to [d]$ of its coordinates. For example, the *input gradient attribution* scheme takes as input instance $x$ & predicted label $\hat{y}$ and outputs an ordering $[d]$ that ranks coordinates in decreasing order of their input gradient magnitude. That is, coordinate $j$ is ranked ahead of coordinate $k$ if the magnitude of the $j^{\text{th}}$ coordinate of $\nabla_x \text{Logit}_\theta(x, \hat{y})$ is larger than that of the $k^{\text{th}}$ coordinate.

**Unmasking schemes**. Given instance $x$ and a subset $S \subseteq [d]$ of coordinates, the *unmasked* instance $x^S$ zeroes out all coordinates that are not in subset $S$: $x_j^S = x_j$ if $j \in S$ and 0 if $j \notin S$. An *unmasking scheme* $A : \mathbb{R}^d \to \{S : S \subseteq [d]\}$ simply maps instance $x$ to a subset $A(x) \subseteq [d]$ of coordinates that can be used to obtain unmasked instance $x^{A(x)}$. Any attribution scheme $\mathcal{A}$ naturally induces *top-k* and *bottom-k* unmasking schemes, $\mathcal{A}_k^{\text{top}}$ and $\mathcal{A}_k^{\text{bot}}$, which output $k$ coordinates with the top-most and bottom-most attributions in $\mathcal{A}(x)$ respectively. In other words, given attribution scheme $\mathcal{A}$ and level $k$, the top-k and bottom-k unmasking schemes, $\mathcal{A}_k^{\text{top}}$ and $\mathcal{A}_k^{\text{bot}}$, can be defined as follows:

$$\mathcal{A}_k^{\text{top}}(x) := \{\mathcal{A}(x)_j : j \leq k\},$$
$$\mathcal{A}_k^{\text{bot}}(x) := \{\mathcal{A}(x)_j : d - k < j \leq d\}.$$

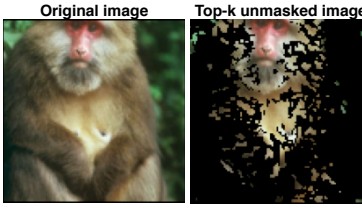
Original image    Top-k unmasked image

For example, Figure 2 depicts an image $x$ and its top-$k$ unmasked variant $x^{\mathcal{A}_k^{\text{top}}(x)}$. In this case, the attribution scheme $\mathcal{A}$ assigns higher rank to pixels in the foreground. So, the top-25% unmasking operation, $x^{\mathcal{A}_{25\%}^{\text{top}}(x)}$, highlights the monkey by retaining pixels with top-25% attribution ranks and zeroing out the remaining pixels that correspond to the green background.

Figure 2: Pictorial example of a top-25% unmasked image.

**Predictive power of unmasking schemes**. The *predictive power* of an unmasking scheme $A$ with respect to model architecture $M$ (e.g., resnet18) can be defined as the best classification accuracy that can be attained by training a model with architecture $M$ on unmasked instances that are obtained via unmasking scheme $A$. More formally, it can defined as follows:

$$\text{PredPower}_M(A) := \sup_{f \in M, f : \mathbb{R}^d \to \mathcal{Y}} \mathbb{E}_{\mathcal{D}}\left[\mathbb{1}\left[f(x^{A(x)}) = y\right]\right].$$

Due to masking-induced distribution shifts, models with architecture $M$ that are trained using original data cannot be plugged in to estimate $\text{PredPower}_M(A)$. The `ROAR` framework [14] sidesteps this issue by *retraining* models on unmasked data, as similar model architectures tend to learn "similar" classifiers [34, 35, 36, 37]. Therefore, we employ the `ROAR` framework to estimate $\text{PredPower}_M(A)$ in two steps. First, we use unmasking scheme $A$ to obtain *unmasked* train and test datasets that comprise data points of the form $(x^{A(x)}, y)$. Then, we *retrain* a new model with the same architecture $M$ on unmasked train data and evaluate its accuracy on unmasked test data.

`DiffROAR` **evaluation metric to test assumption (A)**. Recall that an attribution scheme $\mathcal{A}$ maps an instance $x$ to a permutation of its coordinates that reflects the order of *estimated* importance in model prediction. An attribution scheme that satisfies assumption (A) must place coordinates that are more important for model prediction higher up in the the attribution order. More formally, given attribution scheme $\mathcal{A}$, architecture $M$ and level $k$, we define `DiffROAR` as the difference between the predictive power of top-$k$ and bottom-$k$ unmasking schemes, $\mathcal{A}_k^{\text{top}}$ and $\mathcal{A}_k^{\text{bot}}$:

$$\text{DiffROAR}_M(\mathcal{A}, k) = \text{PredPower}_M(\mathcal{A}_k^{\text{top}}) - \text{PredPower}_M(\mathcal{A}_k^{\text{bot}}) \tag{1}$$

**Interpreting the `DiffROAR` metric**. The sign of the `DiffROAR` metric indicates whether the given attribution scheme satisfies or violates assumption (A). For example, $\text{DiffROAR}_M(\mathcal{A}, \cdot) < 0$ implies that $\mathcal{A}$ violates assumption (A) , as coordinates with *higher* attribution ranks have *worse* predictive power with respect to architecture $M$. Similarly, the magnitude of the `DiffROAR` metric quantifies the extent to which the ordering in attribution scheme $\mathcal{A}$ separates the most and least discriminative coordinates into two disjoint subsets. For example, a *random* attribution scheme $\mathcal{A}_r$, which outputs attributions $\mathcal{A}_r(x)$ chosen uniformly at random from all permutations of $[d]$, neither highlights nor suppresses discriminative features; $\mathbb{E}[\text{DiffROAR}_M(\mathcal{A}_r, k)] = 0$ for any architecture $M$.

**On testing assumption (A)**. To verify (A) for a given attribution scheme $\mathcal{A}$, it is necessary to evaluate whether input coordinates with *higher* attribution rank are *more* important for model prediction than coordinates with *lower* rank. Consequently, the `ROAR`-based metric in Hooker et al. [14], which essentially computes the top-$k$ predictive power, is not sufficient to test whether attribution methods satisfy assumption (A). Therefore, as discussed above, `DiffROAR` tests (A) by comparing the top-$k$ predictive power, $\text{PredPower}_M(\mathcal{A}_k^{\text{top}})$, to the bottom-$k$ predictive power, $\text{PredPower}_M(\mathcal{A}_k^{\text{bot}})$, using multiple values of $k$.

## 4  Testing assumption (A) on image classification benchmarks

In this section, we use `DiffROAR` to evaluate whether input gradient attributions of standard and adversarially robust MLPs and CNNs trained on four image classification benchmarks satisfy assumption (A). We first summarize the experiment setup and then describe key empirical findings.

**Datasets and models**. We consider four benchmark image classification datasets: SVHN [38], Fashion MNIST [39], CIFAR-10 [40] and ImageNet-10 [41]. ImageNet-10 is an open-sourced variant (`https://github.com/MadryLab/robustness/`) of Imagenet [41], with $80,000$ images grouped into 10 super-classes. ImageNet-10 enables us to test assumption (A) on Imagenet without the computational overload of training models on the 1000-way ILSVRC classification task [42]. We evaluate input gradient attributions of standard and adversarially trained two-hidden-layer MLPs and Resnets [43]. We obtain $\ell_2$ and $\ell_\infty$ $\epsilon$-robust models with perturbation budget $\epsilon$ using PGD adversarial training [15]. Unless mentioned otherwise, we train models using stochastic gradient descent (SGD), with momentum 0.9, batch size 256, $\ell_2$ regularization 0.0005 and initial learning rate 0.1 that decays by a factor of 0.75 every 20 epochs. Additionally, we use standard data augmentation and train models for at most 500 epochs, stopping early if cross-entropy loss on training data goes below 0.001. Appendix C.1 provides additional details about the datasets and trained models.[3]

**Estimating `DiffROAR` on real data**. We compute the evaluation metric, $\text{DiffROAR}_M(\mathcal{A}, k)$, on real datasets in four steps, as follows. First, we train a standard or robust model with architecture $M$ on the original dataset and obtain its input gradient attribution scheme $\mathcal{A}$. Second, as outlined in Section 3, we use attribution scheme $\mathcal{A}$ and level $k$ (i.e., fraction of pixels to be unmasked) to extract the top-$k$ and bottom-$k$ unmasking schemes: $\mathcal{A}_k^{\text{top}}$ and $\mathcal{A}_k^{\text{bot}}$. Third, we apply $\mathcal{A}_k^{\text{top}}$ and $\mathcal{A}_k^{\text{bot}}$ on the original train & test datasets to obtain top-$k$ and bottom-$k$ unmasked datasets respectively. Finally, to compute $\text{DiffROAR}_M(\mathcal{A}, k)$ via eq. (1), we estimate top-$k$ and bottom-$k$ predictive power, $\text{PredPower}_M(\mathcal{A}_k^{\text{top}})$ and $\text{PredPower}_M(\mathcal{A}_k^{\text{bot}})$, by *retraining new models* with architecture $M$ on top-$k$ and bottom-$k$ unmasked datasets respectively. Also, note that we (a) average the `DiffROAR` metric over five runs for each model and unmasking fraction or level $k$ and (b) unmask individual image pixels without grouping them channel-wise.

**Experiment setup**. Now, we analyze the `DiffROAR` metric as a function of the unmasking fraction $k \in \{5, 10, 20, \ldots, 100\}\%$ in order to evaluate whether input gradient attributions of models trained on four image classification benchmarks satisfy assumption (A). In particular, as shown in Figure 3, we use `DiffROAR` to analyze input gradients of standard and adversarially robust two-hidden-layer MLPs on SVHN & Fashion MNIST, Resnet18 on ImageNet-10, and Resnet50 on CIFAR-10. In order to calibrate our findings, we compare input gradient attributions of these models to two natural baselines: model-agnostic *random* attributions and input-agnostic attributions of linear models.

**Input gradients of standard models**. Input gradient attributions of standard MLPs trained on SVHN satisfy assumption (A), as the `DiffROAR` metric in Figure 3(a) is positive for all values of level $k < 100\%$. However, in Figure 3(b), the `DiffROAR` curves of standard MLPs trained on

---

[3]Code publicly available at `https://github.com/harshays/inputgradients`

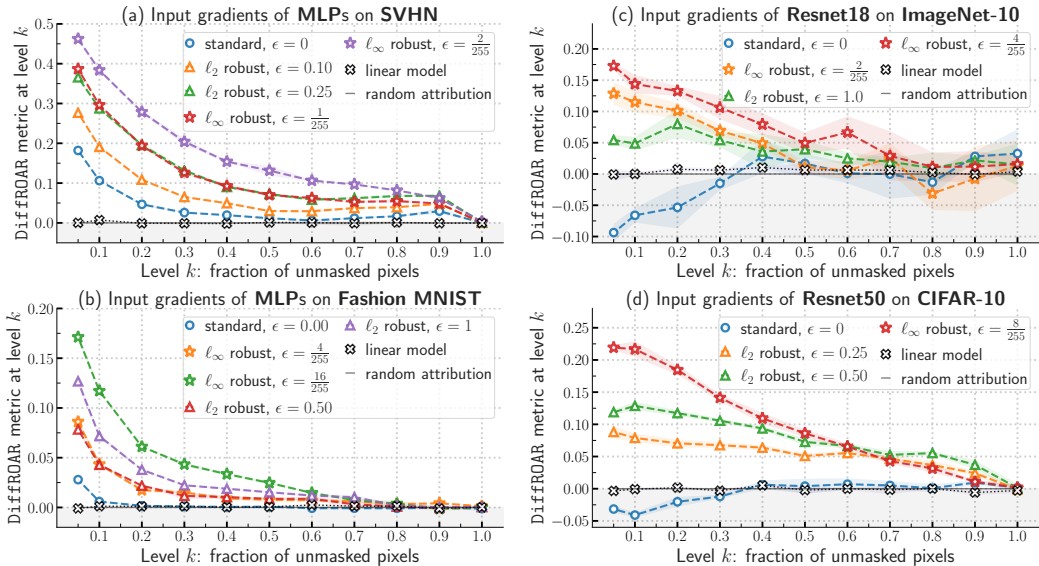

Figure 3: `DiffROAR` plots for input gradient attributions of standard and adversarially robust two-hidden-layer MLPs on (a) SVHN & (b) Fashion MNIST, (c) Resnet18 on ImageNet-10 and (d) Resnet50 on CIFAR-10. Subplot (a) indicates that adversarially robust MLPs consistently and considerably outperform standard MLPs on the `DiffROAR` metric for all $k < 100\%$. Subplot (b) shows that for most unmasking fractions $k$, standard MLPs trained on Fashion MNIST, unlike robust MLPs, fare no better than model-agnostic random attributions and input-agnostic attributions of linear models. Subplots (c) and (d) show that when $k < 40\%$, standard Resnet models trained on CIFAR-10 and ImageNet-10 grossly violate (A), thereby implying that coordinates with top-most gradient attribution rank have worse predictive power than coordinates with bottom-most rank. In stark contrast, input gradients of Resnets that are robust to $\ell_2$ and $\ell_\infty$ adversarial perturbations satisfy assumption (A) reasonably well. We observe that increasing the perturbation budget $\epsilon$ during adversarial training amplifies the magnitude of `DiffROAR` for every $k$ across all four image classification benchmarks.

Fashion MNIST indicate that input gradient attributions, consistent with findings in Hooker et al. [14], can fare no better than model-agnostic random attributions and input-agnostic attributions of linear models vis-a-vis assumption (A). Furthermore, and rather surprisingly, the shaded area in Figure 3(c) and Figure 3(d) shows that when level $k < 40\%$, `DiffROAR` curves of standard Resnets trained on CIFAR-10 and Imagenet-10 are consistently *negative* and perform considerably worse than model-agnostic and input-agnostic baseline attributions. These results strongly suggest that on CIFAR-10 and Imagenet-10, input gradients of standard Resnets grossly violate assumption (A) and suppress discriminative features. In other words, coordinates with *larger* gradient magnitude have *worse* predictive power than coordinates with *smaller* gradient magnitude.

**Input gradients of robust models**. Models that are $\epsilon$-robust to $\ell_2$ and $\ell_\infty$ adversarial perturbations fare considerably better than standard models on the `DiffROAR` metric. For example, in Figure 3(a), when level $k$ equals 10%, robust MLPs trained on SVHN outperform standard MLPs on the `DiffROAR` metric by roughly 10-30%. The `DiffROAR` curves of adversarially robust MLPs in Figure 3 are positive at every level $k < 100\%$, which strongly suggests that input gradient attributions of robust MLPs satisfy assumption (A). Similarly, robust resnet50 models trained on CIFAR-10 and ImageNet-10 satisfy assumption (A) reasonably well and, unlike standard resnet50 models, starkly highlight discriminative features. Furthermore, we observe that increasing the perturbation budget $\epsilon$ in $\ell_2$ or $\ell_\infty$ PGD adversarial training [15] amplifies the magnitude of `DiffROAR` across $k$ and for all four datasets. That is, the adversarial perturbation budget $\epsilon$ determines the extent to which input gradients differentiates the most and least discriminative coordinates into two disjoint subsets.

**Additional results**. In Appendix C, we show that our `DiffROAR` results are robust to choice of model architecture & SGD hyperparameters during retraining and also hold for input gradients taken with respect to cross-entropy. Additionally, while `DiffROAR` *without retraining* gives qualitatively similar results, they are not as consistent across architectures as with retraining, particularly for small unmasking fraction $k$ that induce non-trivial distribution shifts.

# 5    Analyzing input gradient attributions using `BlockMNIST` data

To verify whether input gradients satisfy assumption (`A`) more thoroughly, we introduce and perform experiments on `BlockMNIST`, an `MNIST`-based dataset that *by design* encodes a priori knowledge of ground-truth discriminative features.

`BlockMNIST` **dataset design**: The design of the `BlockMNIST` dataset is based on two intuitive properties of real-world object classification tasks: (i) for different images, the object of interest may appear in different parts of the image (e.g., top-left, bottom-right); (ii) the object of interest and the rest of the image often share low-level patterns such as edges that are not informative of the label on their own. We replicate these aspects in `BlockMNIST` instances, which are vertical concatenations of two $28 \times 28$ *signal* and *null* image blocks that are randomly placed at the top or bottom with equal probability. The *signal* block is an `MNIST` image of digit $0$ or digit $1$, corresponding to class $0$ or $1$ of the `BlockMNIST` image respectively. On the other hand, the *null* block in every `BlockMNIST` image, independent of its class, contains a square patch made of two horizontal, vertical, and slanted lines, as shown in Figure 1(a). It is important to note that unlike the `MNIST` signal block that is fully predictive of the class, the non-discriminative null block contains no information about the class. Standard as well as adversarially robust models trained on `BlockMNIST` data attain 99.99% test accuracy, thereby implying that model predictions are indeed based solely on the signal block for any given instance. We further verify this by noting that the predictions of trained model remain unchanged on almost every instance even when all pixels in the null block are set to zero.

**Do standard and robust models satisfy (`A`)?** As discussed above, unlike the null block that has no task-relevant information, the `MNIST` digit in the signal block entirely determines the class of any given `BlockMNIST` image. Therefore, in this setting, we can restate assumption (`A`) as follows: *Do input gradient attributions highlight the signal block over the null block?* Surprisingly, as shown in Figure 1(b,c), input gradient attributions of standard MLP and Resnet18 models highlight the signal block *as well as* the non-discriminative null block. In stark contrast, subplots (d) and (e) in Figure 1 show that input gradient attributions of $\ell_2$ robust MLP and Resnet18 models exclusively highlight `MNIST` digits in the signal block and clearly suppress the square patch in the null block. These results validate our findings on real-world datasets by showing that unlike standard models, adversarially robust models satisfy (`A`) on `BlockMNIST` data.

**Feature leakage hypothesis**: Recall that the discriminative signal block in `BlockMNIST` images is randomly placed at the top or bottom with equal probability. Given our results in Figure 1, we hypothesize that when discriminative features vary across instances (e.g., signal block at top vs. bottom), input gradients of standard models not only highlight instance-specific features but also *leak* discriminative features from other instances. We term this hypothesis *feature leakage*.

To test our hypothesis, we leverage the modular structure in `BlockMNIST` to construct a slightly modified version, `BlockMNIST-Top`, wherein the location of the `MNIST` signal block is fixed at the top for all instances (see fig. 4). In this setting, in contrast to results on `BlockMNIST`, input gradients of *standard* Resnet18 and MLP models trained on `BlockMNIST-Top` satisfy assumption (`A`). Specifically, when the signal block is fixed at the top, input gradient attributions in Figure 4(b, c) clearly highlight the signal block and suppress

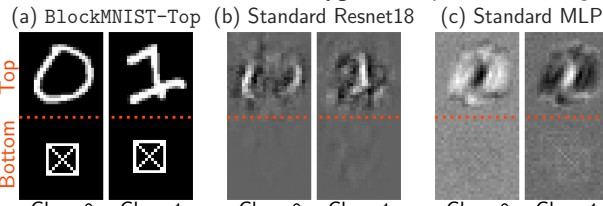

Figure 4: (a) In `BlockMNIST-Top` images, the signal & null blocks are fixed at the top & bottom respectively. In contrast to results on `BlockMNIST` in fig. 1, input gradients of standard (b) Resnet18 and (c) MLP trained on `BlockMNIST-Top` highlight discriminative features in the signal block, suppress the null block, and satisfy (`A`).

the null block, thereby supporting our feature leakage hypothesis. Based on our `BlockMNIST` experiments, we believe that understanding *how* adversarial robustness mitigates feature leakage is an interesting direction for future work.

**Additional results**. In Appendix D.1, we (i) visualize input gradients of several `BlockMNIST` and `BlockMNIST-Top` images, (ii) introduce a quantitative proxy metric to compare feature leakage between standard and robust models, (iii) show that our findings are fairly robust to the choice and number of classes in `BlockMNIST` data, and (iv) evaluate feature leakage in five feature attribution methods. We also provide experiments that falsify hypotheses vis-a-vis input gradients and assumption (`A`) that we considered in addition to feature leakage.

# 6    Feature leakage in input gradient attributions

To understand the extent of feature leakage more thoroughly, we introduce a simplified version of the `BlockMNIST` dataset that is amenable to theoretical analysis. We rigorously show that input gradients of standard one-hidden-layer MLPs do not differentiate instance-specific features from other task-relevant features that are not pertinent to the given instance.

**Dataset**: Given dimension of each block $\widetilde{d}$, feature vector $u^* \in \mathbb{R}^{\widetilde{d}}$ with $\|u^*\| = 1$, number of blocks $d$ and noise parameter $\eta$, we will construct input instances of dimension $\widetilde{d} \cdot d$. More concretely, a sample $(x, y) \in \mathbb{R}^{\widetilde{d} \cdot d} \times \{\pm 1\}$ from the distribution $\mathcal{D}$ is generated as follows:

$y = \pm 1$ with probability 0.5 and

$$x = [\eta g_1, \quad \eta g_2, \quad \dots, \quad yu^* + \eta g_j, \quad \dots, \quad \eta g_d] \text{ with } j \text{ chosen at random from } [d/2] \quad (2)$$

where each $g_i \in \mathbb{R}^{\widetilde{d}}$ is drawn uniformly at random from the unit ball. For simplicity, we take $d$ to be even so that $d/2$ is an integer. We can think of each $x$ as a concatenation of $d$ $\widetilde{d}$-dimensional blocks $\{x_1, \dots, x_d\}$. The first $d/2$ blocks, $\{1, \dots, d/2\}$, are *task-relevant*, as every example $(x, y)$ contains an instance-specific *signal* block $x_i = yu^* + \eta g_i$ for some $i \in [d/2]$ that is informative of its label $y$. Given instance $x$, we use $j^*(x)$ to denote the unique instance-specific signal block such that $x_{j^*(x)} = yu^* + \eta g_{j^*(x)}$. On the other hand, *noise* blocks $\{d/2 + 1, \dots, d\}$ do not contain task-relevant signal for any instance $x$. At a high level, the instance-specific signal block $j^*(x)$ and noise blocks $\{d/2 + 1, \dots, d\}$ in instance $x$ correspond to the discriminative `MNIST` digit and the null square patch in `BlockMNIST` images respectively. For example, each row in Figure 5(a) illustrates an instance $x$ where $d = 10, \widetilde{d} = 1, \eta = 0$ and $u^* = 1$.

**Model**: We consider one-hidden layer MLPs with ReLU nonlinearity in the infinite-width limit. More concretely, for a given width $m$, the network is parameterized by $R \in \mathbb{R}^{m \times \widetilde{d} \cdot d}, b \in \mathbb{R}^m$ and $w \in \mathbb{R}^m$. Given an input instance $(x, y) \in \mathbb{R}^{\widetilde{d}d} \times \{\pm 1\}$, the output score (or logit) $f$ and cross-entropy (CE) loss $\mathcal{L}$ are given by:

$$f((w, R, b), x) := \langle w, \phi(Rx + b) \rangle, \quad \mathcal{L}((w, R, b), (x, y)) := \log(1 + \exp(-y \cdot f((w, R, b), x))).$$

where $\phi(t) := \max(0, t)$ denotes the ReLU function. A remarkable set of recent results [44, 45, 46, 47] show that as $m \to \infty$, the training procedure is equivalent to gradient descent (GD) on an infinite dimensional Wasserstein space. In the Wasserstein space, the network can be interpreted as a probability distribution $\nu$ over $\mathbb{R} \times \mathbb{R}^{\widetilde{d} \cdot d} \times \mathbb{R}$ with output score $f$ and cross entropy loss $\mathcal{L}$ defined as:

$$f(\nu, x) := \mathbb{E}_{(w, r, b) \sim \nu}[w\phi(\langle r, x \rangle + b)], \quad \mathcal{L}(\nu, (x, y)) := \log(1 + \exp(-y \cdot f(\nu, x))). \quad (3)$$

**Theoretical analysis**: Our approach leverages the recent result in Chizat and Bach [48], which shows that if GD in the Wasserstein space $\mathcal{W}^2\left(\mathbb{R} \times \mathbb{R}^{\widetilde{d}d} \times \mathbb{R}\right)$ on $\mathbb{E}_{\mathcal{D}}[\mathcal{L}(\nu, (x, y))]$ converges, it does so to a max-margin classifier given by:

$$\nu^* := \underset{\nu \in \mathcal{P}\left(\mathbb{S}^{d\widetilde{d}+1}\right)}{\arg\max} \underset{(x, y) \sim \mathcal{D}}{\min} y \cdot f(\nu, x), \quad (4)$$

where $\mathbb{S}^{d\widetilde{d}+1}$ denotes the surface of the Euclidean unit ball in $\mathbb{R}^{d\widetilde{d}+2}$, and $\mathcal{P}\left(\mathbb{S}^{d\widetilde{d}+1}\right)$ denotes the space of probability distributions over $\mathbb{S}^{d\widetilde{d}+1}$. Intuitively, our main result shows that on any data point $(x, y) \sim \mathcal{D}$, the input gradient magnitude of the max-margin classifier $\nu^*$ is *equal* over all task-relevant blocks $\{1, \dots, d/2\}$ and zero on the remaining *noise* blocks $\{d/2 + 1, \dots, d\}$.

**Theorem 1.** *Consider distribution $\mathcal{D}$ (2) with $\eta < \frac{1}{10d}$. There exists a max-margin classifier $\nu^*$ for $\mathcal{D}$ in Wasserstein space (i.e., training both layers of FCN with $m \to \infty$) given by (4), such that for all $\forall (x, y) \sim \mathcal{D}$: (i) $\left\|(\nabla_x \mathcal{L}(\nu^*, (x, y)))_j\right\| = c > 0$ for every $j \in [d/2]$ and (ii) $\left\|(\nabla_x \mathcal{L}(\nu^*, (x, y)))_j\right\| = 0$ for every $j \in \{d/2 + 1, \cdots, d\}$, where $(\nabla_x \mathcal{L}(\nu^*, (x, y)))_j$ denotes the $j^{th}$ block of the input gradient $\nabla_x \mathcal{L}(\nu^*, (x, y))$.*

Theorem 1 guarantees the *existence* of a max-margin classifier such that the input gradient magnitude for any given instance is (i) a non-zero constant on each of the first $d/2$ task-relevant blocks, and

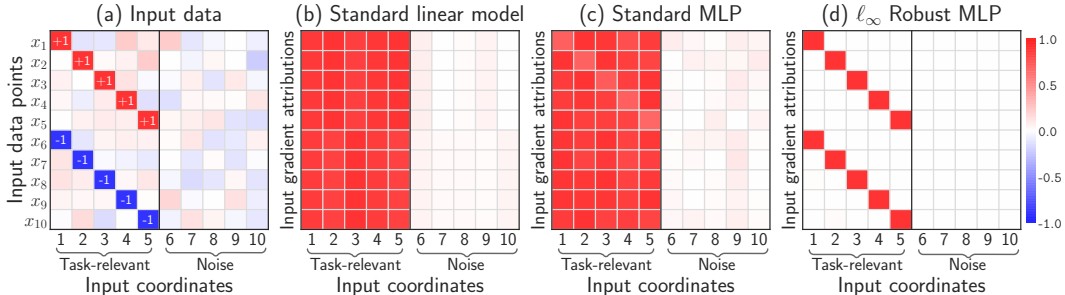

Figure 5: Input gradients of linear models and standard & robust MLPs trained on data from eq. (2) with $d = 10, \widetilde{d} = 1, \eta = 0$ and $u^* = 1$. (a) Each row in corresponds to an instance $x$, and the highlighted coordinate denotes the signal block $j^*(x)$ & label $y$. (b) Linear models suppress noise coordinates but lack the expressive power to highlight instance-specific signal $j^*(x)$, as their input gradients in subplot (b) are identical across all examples. (c) Despite the expressive power to highlight instance-specific signal coordinate $j^*(x)$, input gradients of standard MLPs exhibit feature leakage (see Theorem 1) and violate (A) as well. (d) In stark contrast, input gradients of adversarially trained MLPs suppress feature leakage and starkly highlight instance-specific signal coordinates $j^*(x)$.

(ii) equal to zero on the remaining $d/2$ *noise* blocks that do not contain any information about the label. However, input gradients fail at highlighting the *unique instance-specific signal* block over the remaining *task-relevant* blocks. This clearly demonstrates feature leakage, as input gradients for any given instance also highlight task-relevant features that are, in fact, *not specific* to the given instance. Therefore, input gradients of standard one-hidden-layer MLPs do not highlight instance-specific discriminative features and grossly violate assumption (A). In Appendix F, we present additional results that demonstrate that adversarially trained one-hidden-layer MLPs can suppress feature leakage and satisfy assumption (A).

**Empirical results**: Now, we supplement our theoretical results by evaluating input gradients of linear models as well as standard & robust one-hidden-layer ReLU MLPs with width $10000$ on the dataset shown in Figure 5. Note that all models obtain 100% test accuracy on this linearly separable dataset, a simplified version of BlockMNIST that is obtained via eq. 2 with $d = 10, \widetilde{d} = 1, \eta = 0$ and $u^* = 1$. Due to insufficient expressive power, linear models have input-agnostic gradients that suppress all five noise coordinates, but do not differentiate the instance-specific signal coordinate from the remaining task-relevant coordinates. Consistent with Theorem 1, even standard MLPs, which are expressive enough to have input gradients that correctly highlight instance-specific coordinates, apply equal weight on all five task-relevant coordinates and violate (A) due to feature leakage. On the other hand, Figure 5(c) shows that the same MLP architecture, if robust to $\ell_\infty$ adversarial perturbations with norm $0.35$, satisfies (A) by clearly highlighting the instance-specific signal coordinate over all other noise *and* task-relevant coordinates

## 7 Discussion and conclusion

In this work, we took a three-pronged approach to investigate the validity of a key assumption made in several popular post-hoc attribution methods: (A) *coordinates with larger input gradient magnitude are more relevant for model prediction compared to coordinates with smaller input gradient magnitude*. Through (i) evaluation on real-world data using our DiffROAR framework, (ii) empirical analysis on BlockMNIST data that encodes information of ground-truth discriminative features, and (iii) a rigorous theoretical study, we present strong evidence to suggest that standard models do not satisfy assumption (A). In contrast, adversarially robust models satisfy (A) in a consistent manner. Furthermore, our analysis in Section 5 and Section 6 indicates that *feature leakage* sheds light on why input gradients of standard models tend to violate (A). We provide additional discussion in Appendix B.

This work exclusively focused on "vanilla" input gradients due to their fundamental significance in *feature attribution*. A similarly thorough investigation that analyzes other commonly-used attribution methods is an interesting avenue for future work. Another interesting avenue for further analyses is to understand how adversarial training mitigates feature leakage in input gradient attributions.

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
