# Appendices

The supplementary material is organized as follows. We first discuss additional related work Section 2. Appendix B provides additional discussion. Appendix C describes additional experiments based on the `DiffROAR` framework to analyze the fidelity of input gradient attributions on real-world datasets. In Appendix D, we provide additional experiments on feature leakage using `BlockMNIST`-based datasets. Then, Appendix E contains the proof of Theorem 1 and Appendix F discusses the effect of adversarial training on input gradients of models that are adversarially trained on a simplified version of `BlockMNIST` data. We plan to open-source our trained models, code primitives, and Jupyter notebooks soon, which can be used to reproduce our empirical results.

## A  Additional related work

In this section, we briefly describe works that analyze two properties of post-hoc instance-specific explanations that are related to explanation fidelity or "correctness". In particular, we outline recent works that study the *robustness* and *practical utility* of instance-specific explanation methods.

**Robustness of explanations**: Several commonly used instance-specific explanation methods lack robustness in practice. Ghorbani et al. [49] show that instance-specific explanations and exemplar-based explanations are not robust to imperceptibly small adversarial perturbations to the input. Heo et al. [50] show that instance-specific explanations are highly vulnerable to adversarial *model* manipulations as well. Dombrowski et al. [51] show that explanations lack robustness to to *arbitrary* manipulations and show that non-robustness stems from geometric properties of neural networks. Bansal et al. [52] show that explanation methods are considerably sensitive to method-specific hyperparameters such as sample size, blur radius, and random seeds. Recent works promote robustness in explanations using smoothing [51] or variants of adversarial training [53, 54]

**Utility of explanations**: A recent line of work propose evaluation frameworks to assess the practical utility of post-hoc instance-specific explanation methods via proxy downstream tasks. Chu et al. [55] employ a randomized controlled trial to show that using explanation methods as additional information does not improve *human* accuracy on classification tasks. More generally, Poursabzi-Sangdeh et al. [56] analyze the effect of model transparency (e.g., number of input features, black-box vs. white-box) on the accuracy of human decisions with respect to the task and model. Similarly, Adebayo et al. [5] conduct a human subject study to show that subjects fail to identify defective models using attributions and instead primarily rely on model predictions. [57] formalize the "value" of explanations as the explanation utility (i.e., as side information) in a student-teacher learning framework. In contrast to the works above, we propose an evaluation framework, `DiffROAR`, to evaluate the fidelity, or "correctness", of explanations in classification tasks. In particular, using benchmark image classification tasks and synthetic data, we empirically and theoretically characterize input gradient attributions of standard as well as adversarially robust models.

**Stability of explanations**. Explanation stability and explanation correctness (also known as explanation fidelity) are two distinct desirable properties of explanations [20]. That is, stability does not imply fidelity. For example, an input-agnostic constant explanation is stable but lacks fidelity. Conversely, fidelity does not imply stability—if the underlying model is itself unstable, then any correct high-fidelity explanation of that model must also be unstable. Bansal et al. [52] and Chen et al. [58] identify and explain why input gradients of adversarially trained models are more stable compared to those of standard models. In contrast, our work focuses on identifying and explaining why input gradients of adversarially trained models have more fidelity compared to those of standard models. Furthermore, we also take the first step towards theoretically showing that adversarial robustness can provably improve input gradient fidelity in Appendix E.

# B  Additional discussion

**Translation invariance in `BlockMNIST` models**. Intuitively, CNNs are translation-invariant only if the object of interest is not closer to the boundary than the receptive field of the final layer; In `BlockMNIST`, the digits are either close to the top boundary or the bottom boundary. Given that the receptive field of Resnets is quite large, translation invariance would not hold in this case. This is further supported by recent work [59], which demonstrates that "CNNs can and will exploit the absolute spatial location by learning filters that respond exclusively to particular absolute locations by exploiting image boundary effects". We observe this phenomenon empirically in our `BlockMNIST-Top` experiments as well. That is, while models trained on BlockMNIST-Top data (i.e., MNIST digit in top block) attain 100% test accuracy on `BlockMNIST-Top` images, the accuracy of these models degrades to approximately 55% (i.e., 5% better than random chance) when evaluated on `BlockMNIST-Bottom` images, wherein the MNIST digit (signal) is placed in the bottom block.

**Choice of removal operator in `DiffROAR` framework**. Recall that in `DiffROAR`, the predictive power of a new model retrained on the unmasked dataset (i.e, data points after removal operation) is used to evaluate the fidelity of post-hoc explanation methods. Note that this approach employs retraining to account for and nullify distribution shifts induced by feature removal operators such as gaussian noise, zeros etc. Since the same removal operation is applied to unmask every image (across classes), the choice of removal operator has no effect on our `DiffROAR` results in Section 4. To verify this, we evaluated `DiffROAR` on CIFAR-10 with another removal operator in which pixels are masked/replaced by random gaussian noise (instead of zeros) and observed that the results do not change (i.e., same as in Figure 3).

**Counterfactual changes vis-a-vis feature leakage**. As evidenced in the `BlockMNIST` experiments, input gradient attributions of standard models incorporate counterfactual changes in the null block. While this phenomenon seems natural and "intuitive" in hindsight, it can be misleading in the context of feature attributions. For example, consider the typical use case for feature attributions: to highlight regions within the given instance/image that are most relevant for model prediction. Now, in the `BlockMNIST` setting, if input gradients leak digit-like features into the null block, then the feature attributions in the null block can be easily (mis)interpreted as the non-discriminative null patch being highly relevant for model prediction.

**Comparison to results in Kim et al. [33]**. Kim et al. [33] use the `ROAR` framework to conjecture that adversarial training "tilts" input gradients to better align with the data manifold. First, in contrast to Kim et al. [33], we thoroughly establish our `DiffROAR` results across datasets/architectures/hyperparameters, revealing a significantly larger gap between the attribution quality of standard and adversarially robust models. Second, motivated by the boundary tilting hypothesis [60], Kim et al. [33] use a two-dimensional synthetic dataset to empirically show that the decision boundary of robust models aligns better with the vector between the two class-conditional means. However, this empirical evidence might be misleading, as Ilyas et al. [61] theoretically demonstrates that "this exact statement is not true beyond two dimensions" (pg. 15). Furthermore, several recent works have also provided concrete evidence to support alternative hypotheses [61, 37, 62, 63] for the existence of adversarial examples that counter the boundary tilting hypothesis that Kim et al. build upon. This discrepancy in these results motivates the need for a multipronged approach, which we adopt to empirically identify the feature leakage hypothesis using BlockMNIST and theoretically verify the hypothesis in Section 6.

**Connections between adversarial robustness and data manifold**: In the recent past, there have been several results showing unexpected benefits of adversarially trained models beyond adversarial robustness such as visually perceptible input gradients [29] and feature representations that transfer better [64]. One reason for this phenomenon widely considered in the literature [65, 66] is that the input data lies on a low dimensional manifold and unlike standard training, adversarial training encourages the decision boundary to lie on this manifold (i.e. alignment with data manifold). Our experiments and theoretical results on feature leakage suggest that this reasoning is indeed true for both the `BlockMNIST` and its simplified version presented in Section 6. Furthermore, we believe that the simplified version of `BlockMNIST` in eq. (2) can be used as a tool to thoroughly investigate both the benefits and potential drawbacks of adversarially trained models.

**Why focus on input gradient attributions?**. As discussed in Section 1, several feature attributions such as guided backprop [16] and integrated gradients [22] that output visually sharper saliency maps fail basic sanity checks such as model randomization and label randomization [12, 13, 20]. We focus on vanilla input gradient attributions for two key reasons: (i) vanilla input gradients pass both sanity checks mentioned above and (ii) the input gradient operation is the key building block of several feature attribution methods. Our experiments and theoretical analysis are specifically designed to identify and verify feature leakage in input gradient attributions of standard models.

**Comparing `ROAR` and `DiffROAR`.** The following questions below illustrate key differences between `ROAR` [14] and our work:

- *Does the framework verify assumption (`A`)?* In Hooker et al. [14], the `ROAR` framework essentially computes the top-$k$ predictive power only, which is not sufficient to test assumption (`A`). In our paper, DiffROAR directly compares the top-$k$ and bottom-$k$ predictive power to test whether the given attribution method satisfies assumption (`A`).

- *Are the results in the paper conclusive?* Both, `ROAR` and `DiffROAR`, make a key assumption: models retrained on unmasked datasets learn the same features as the model trained on the original dataset. Although empirically supported [34, 35, 37], this assumption makes it difficult to conclusively test assumption (`A`). Therefore, we empirically (Section 5) and theoretically (Section 6) verify our `DiffROAR` findings in settings wherein ground-truth features are known a priori.

- *Does the work identify why standard input gradients violate (`A`)?* Hooker et al. [14] do not discuss why input gradients lack explanation fidelity. In our paper, we hypothesize feature leakage as the key reason for ineffectiveness of input gradients, and validate it with empirical as well as theoretical analysis on `BlockMNIST`-based data

**Limitations of `ROAR` and `DiffROAR`.** The major limitation of `ROAR` and `DiffROAR` is the key assumption that models retrained on unmasked datasets learn the same features as the model trained on the original dataset. In the absence of ground-truth features, this assumption is empirically supported by findings that suggest that different runs of models sharing the same architecture learn similar features [34, 35, 37]. Another limitation is that `ROAR`-based frameworks are not useful in the following setting. Consider a redundant dataset where features are either all negative (in which case label $y = 0$) or all positive (in which case label $y = 1$). In such cases, no feature is more or less informative than any other, so no information can be gained by ranking or removing input coordinates/features.

## C   Experiments on real-world datasets using `DiffROAR`

In this section, we first provide additional details about datasets, training, and performance of trained models vis-a-vis generalization and robustness. We also present top-$k$ and bottom-$k$ predictive power of input gradient unmasking schemes obtained via standard and robust models. Next, we show that our results on image classification benchmarks are robust to CNN architectures and SGD hyperparameters used during retraining. Then, we use `DiffROAR` to show that our results hold with input *loss* gradients, but *signed* input logit gradients do not satisfy assumption (A) for standard *or* robust models. Finally, we discuss `DiffROAR` results obtained without retraining and provide additional example images that are masked using input gradients of standard & robust models.

### C.1   Additional details about `DiffROAR` experiments and trained models

We first provide additional details about standard and adversarial training, and describe the performance of trained models vis-a-vis generalization and robustness to $\ell_2$ & $\ell_\infty$ perturbations.

Recall that we use `DiffROAR` to analyze input gradients of standard and adversarially robust two-hidden-layer MLPs on SVHN & Fashion MNIST, Resnet18 on ImageNet-10, and Resnet50 on CIFAR-10 in Figure 3. In these experiments, we train models using stochastic gradient descent (SGD), with momentum 0.9, batch size 256, $\ell_2$ regularization 0.0005 and initial learning rate 0.1 that decays by a factor of 0.75 every 20 epochs; We obtain $\ell_2$ and $\ell_\infty$ $\epsilon$-robust models with perturbation budget $\epsilon$ using PGD adversarial training [15]. In PGD adversarial training, we use learning rate $\epsilon/4$, 8 steps of PGD and no random initialization in order to compute $\epsilon$-norm $\ell_2$ and $\ell_\infty$ perturbations. In both cases, we use standard data augmentation and train models for at most 500 epochs, stopping early if cross-entropy (standard or adversarial) loss on training data goes below 0.001. Unless mentioned otherwise, we set the depth and width of MLPs trained on real datasets to be 2 and 2× the input dimension respectively.

Figure 6 depicts standard test accuracy (i.e., when perturbation budget $\epsilon = 0$) and $\epsilon$-robust test accuracy (for multiple values of $\epsilon$) of standard as well as $\ell_2$ and $\ell_\infty$ robust models trained on SVHN, Fashion MNIST, CIFAR-10 and ImageNet-10. Note that to estimate $\epsilon$-robust test accuracy, we use PGD-based adversarial *test* examples, computed using 2× the number of PGD steps used during training. As expected, we observe that (i) compared to standard models, adversarially trained MLPs and CNNs attain significantly better robust test accuracy, (ii) models trained with larger perturbation budget are more robust to larger-norm adversarial perturbations at test time, and (iii) standard test accuracies (when $\epsilon = 0$) of adversarially trained models are worse than those of standard models.

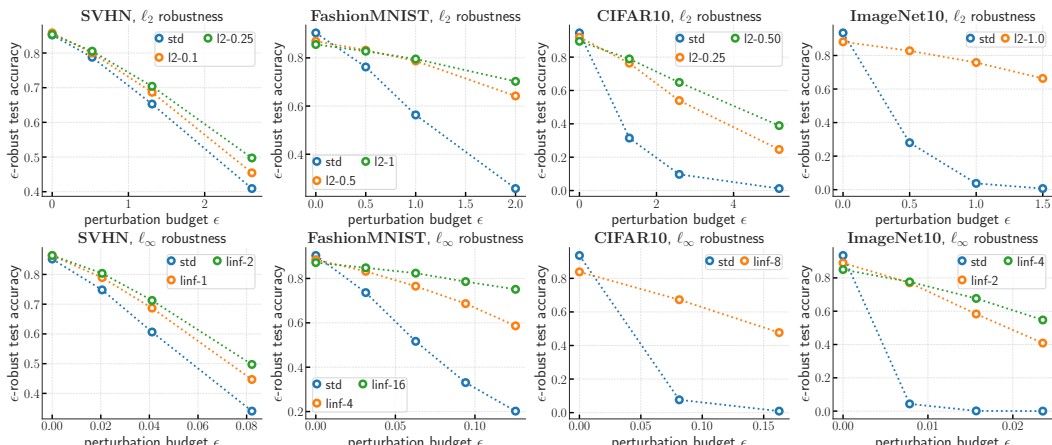

Figure 6: **Standard and $\epsilon$-robust test accuracies** of MLPs trained on SVHN and Fashion MNIST, Resnet50 trained on CIFAR-10, and Resnet18 trained on ImageNet10. Details in Appendix C.1.

### C.2   Top-$k$ and bottom-$k$ predictive power of input gradient attributions

Now, we describe the top-$k$ and bottom-$k$ predictive power curves for unmasking schemes of input gradients of standard and robust models. Recall that top-$k$ predictive power simply estimates the test accuracy of models that are retrained on datasets wherein only coordinates with top-$k$ (%) of

the coordinates are unmasked in every image. The top and bottom rows in Figure 7 show how top-$k$ and bottom-$k$ predictive power of input gradient attributions of standard and robust models vary with unmasking fraction $k$ respectively. The subplots in Figure 7 show that (i) decreasing the unmasking fraction $k$ decreases top-$k$ and bottom-$k$ predictive power, and (ii) models retrained on attribution-masked datasets attain non-trivial unmasked test dataset accuracy even when a significant fraction of coordinates with the top-most and bottom-most attributions are masked.

As described in Section 3, for a given attribution scheme and unmasking fraction or level $k$, DiffROAR (see equation (1)) is positive when the top-$k$ predictive power is greater than the bottom-$k$ predictive power. The subplots in the first column indicate that standard models trained on Fashion MNIST do not satisfy assumption (A) because the top-$k$ and bottom-$k$ unmasking schemes are *equally* ineffective at masking discriminative features. Conversely, the difference between the top-$k$ and bottom-$k$ predictive power of input gradient attributions of robust models is significant. For example, in the second column, for the SVHN model adversarially trained with $\ell_\infty$ perturbations and budget $\epsilon = 2/255$ (purple line), top-$k$ predictive power is roughly $40\%$ more than the bottom-$k$ predictive power when $k = 5\%$. Furthermore, as shown in the third and fourth columns, the top-$k$ and bottom-$k$ curves of standard CNNs trained on CIFAR-10 and ImageNet-10 are "inverted", thereby explaining why DiffROAR is *negative* when unmasking fraction is roughly less than $40\%$.

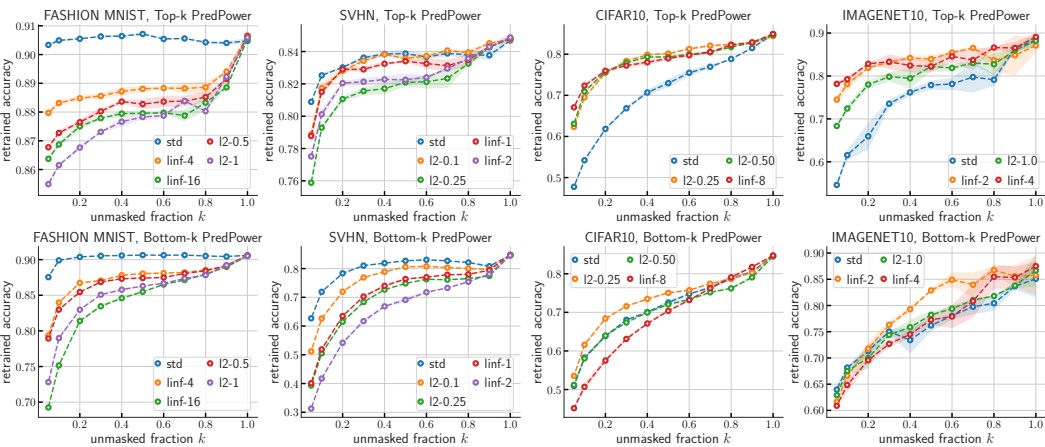

Figure 7: **Predictive power of top-$k$ and bottom-$k$ input gradient unmasking schemes** vs. unmasking fraction, or level, $k$ for standard and adversarially robust models trained on 4 image classification benchmarks. Please see Appendix C.2 for details.

## C.3   Effect of model architecture on DiffROAR results

Recall that in Section 4, we used the DiffROAR metric to evaluate whether input gradient attributions of models trained on real-world datasets satisfy or violate assumption (A). For CNNs, we evaluated input gradient attributions of standard Resnet50 and Resnet18 models trained on CIFAR-10 and Imagenet-10 respectively. In this section, we show that our empirical findings based on these architectures extend to three other commonly-used and well-known CNN architectures: Densenet121, InceptionV3, and VGG11.

As shown in Figure 8, the DiffROAR results support key empirical findings made using input gradients of Resnet models in Section 4: (i) standard models perform poorly, often no better or even worse than the random attribution baseline, and (ii) DiffROAR curves of adversarially robust models are positive and significantly better than that of the standard model. For example, for Densenet121, InceptionV3, and VGG11, when unmasking fraction $k = 20\%$, standard training yields input gradient attributions that attain DiffROAR scores roughly $-5\%$, $2\%$ and $1\%$ respectively, whereas $\ell_\infty$ adversarial training with budget $\epsilon = 6/255$ results in input gradients with DiffROAR metric roughly $15\%$.

## C.4   Effect of SGD Hyperparameters on DiffROAR results

In this section, we show that DiffROAR results for input gradient attribu of standard and robust models are not sensitive to the choice of SGD hyperparameters used during retraining. In particular, we show that DiffROAR curves on CIFAR-10 are not sensitive to the learning rate, weight decay, or the momentum used in SGD to train models on top-$k$ or bottom-$k$ attribution-masked datasets. The

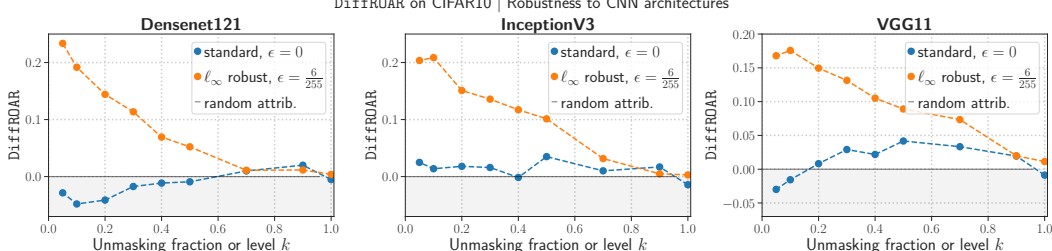

Figure 8: `DiffROAR` **results on input gradients of additional CNN architectures**. `DiffROAR` curves for three well-known NN architectures—Densenet121, InceptionV3, and VGG11—indicate that empirical findings vis-a-vis input gradients of standard and robust models (Section 4) are robust to choice of CNN architecture. Please see Appendix C.3 for details.

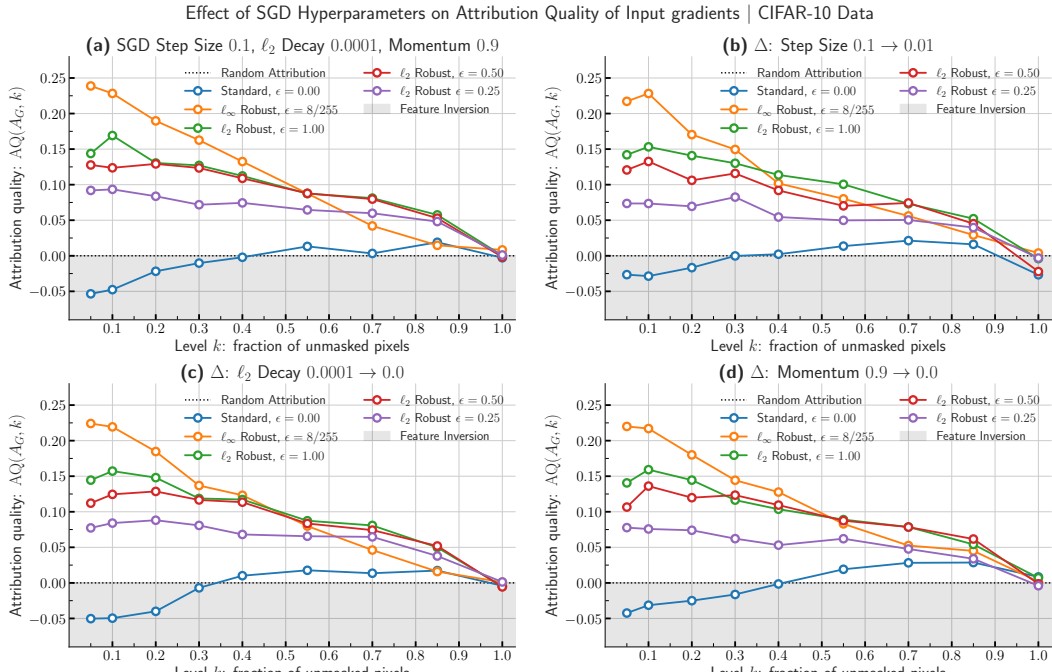

Figure 9: `DiffROAR` **robust to SGD hyperparameters in retraining**. `DiffROAR` curves for input gradients of standard and robust models trained on CIFAR-10 data show that our empirical findings presented in Section 4 are robust to SGD hyperparameters that are used in retraining. Specifically, we show that our findings vis-a-vis `DiffROAR` are not sensitive to changes in SGD hyperparameters such as learning rate, momentum, and weight decay that are used to retrain models on unmasked CIFAR-10 data. For example, the subplots above show that across multiple SGD hyperparameter values, when the fraction of unmasked pixels $k < 30\text{-}40\%$, standard models violate (A) whereas robust models satisfy (A). See Appendix C.4 for details.

four subplots in Figure 9 collectively show that decreasing learning rate from $0.1$ to $0.01$, weight decay from $0.0001$ to $0$, and momentum from $0.9$ to $0$ does not alter our findings: (i) input gradient attributions of standard models do not satisfy (A) when unmasking fraction $k$ is roughly less than 30-40%; (ii) models that are robust to $\ell_2$ and $\ell_\infty$ perturbations consistently satisfy (A); (iii) increasing perturbation budget $\epsilon$ during PGD adversarial training increases `DiffROAR` metric for most values of unmasking fraction $k$. To summarize, our results based on the `DiffROAR` evaluation framework are robust to SGD hyperparameters used to retrain models on top-$k$ and bottom-$k$ unmasked datasets.

## C.5 Evaluating input *loss* gradient attributions using `DiffROAR`

Recall that our experiments in Section 4 evaluate whether input gradients taken w.r.t. the logit of the predicted label satisfy or violate assumption (A) on image classification benchmarks. In this section, we show that our empirical findings generalize to input *loss* gradients—input gradients w.r.t loss

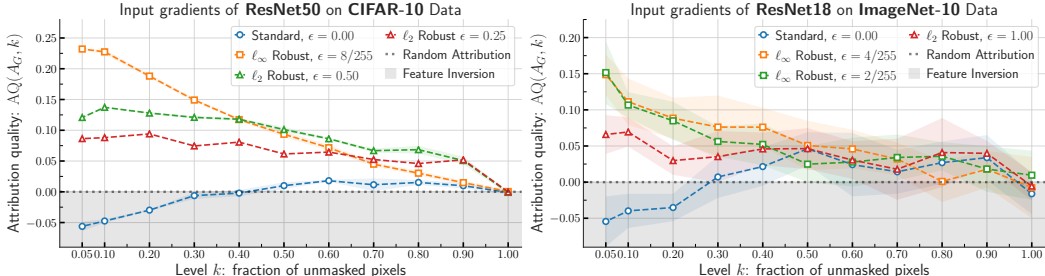

Figure 10: `DiffROAR` **results for input loss gradient attributions**. `DiffROAR` plots for input *loss* gradient attributions of standard and adversarially Resnet50 on CIFAR-10 and Resnet18 on ImageNet-10. In both subplots, standard models violate (A) when the fraction of unmasked pixels $k < 30\%$. That is, input coordinates that have the largest gradient magnitude are not as important performance-wise as the coordinates with smallest gradient magnitude. Conversely, $\{\ell_2, \ell_\infty\}$-adversarially trained models satisfy (A), as the `DiffROAR` metric is positive for all $k < 100\%$. Similar to our results with input logit gradients, we observe that increasing the perturbation budget $\epsilon$ during adversarial training amplifies the magnitude of `DiffROAR` for every $k$ across all four image classification benchmarks.

(e.g., cross-entropy)—of standard and robust models evaluated on image classification benchmarks. Specifically, we apply `DiffROAR` to input *loss* gradients of standard and robust ResNet models trained on CIFAR-10 and ImageNet-10.

Figure 10 illustrates `DiffROAR` curves for input *loss* gradient attributions on CIFAR-10 and ImageNet-10 data. In both cases, we observe that (i) input loss gradient attributions of robust models, unlike those of standard models, satisfy (A) and (ii) PGD adversarial training with larger perturbation budget $\epsilon$ increases the `DiffROAR` metric in a consistent manner. Recall that the magnitude in `DiffROAR` quantifies the extent to which the attribution order separates discriminative and task-relevant features from features that are unimportant for model prediction; see Section 3 for more information about `DiffROAR`.

### C.6 Evaluating *signed* input gradient attributions using `DiffROAR`

In addition to input loss gradient magnitude attributions and input logit gradient magnitude attributions, our results vis-a-vis `DiffROAR` evaluation on image classification benchmarks extend to *signed* input logit gradients as well. In signed input gradient attributions, input coordinates are ranked based on $\text{sgn} x_i \cdot g_i$ where $\text{sgn}(x_i)$ is the sign of input coordinate $x_i$ and $g_i$ is the signed input gradient value for input coordinate $x_i$.

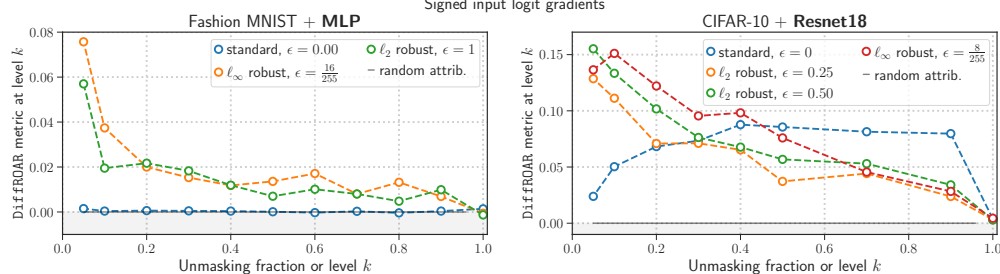

Figure 11: `DiffROAR` **results for signed input logit gradients**. `DiffROAR` results for attributions based on signed input gradients of standard and robust MLPs & CNNs trained on Fashion MNIST & CIFAR-10. See Appendix C.6 for details.

Figure 11 shows `DiffROAR` curves for attributions based on *signed* input gradients taken with respect to the logit of the predicted label. The left and right subplot evaluate `DiffROAR` for standard and robust (i) MLP trained on Fashion MNIST and (ii) Resnet18 models trained on CIFAR-10. Consistent with our findings in Section 4, while standard MLPs trained on Fashion MNIST fare no better than random attributions, signed input gradients of robust MLPs attain positive `DiffROAR` scores for all $k < 100\%$ and perform considerably better than gradients of standard MLPs. Similarly, based on the `DiffROAR` metric, when $k < 50\%$, while signed input gradients of standard Resnet18 models perform better than absolute logit and loss gradients, signed input gradients of robust Resnet18 models continue to fare better than standard models.

## C.7 The role of retraining in `DiffROAR` evaluation

Figure 12 shows the results on `DiffROAR` without retraining on the masked datasets. As we can see from the figures, the trends are not consistent across model architectures and datasets, possibly due to varying levels of distribution shift. For this reason, we employ `DiffROAR` with retraining as described in Section 3.

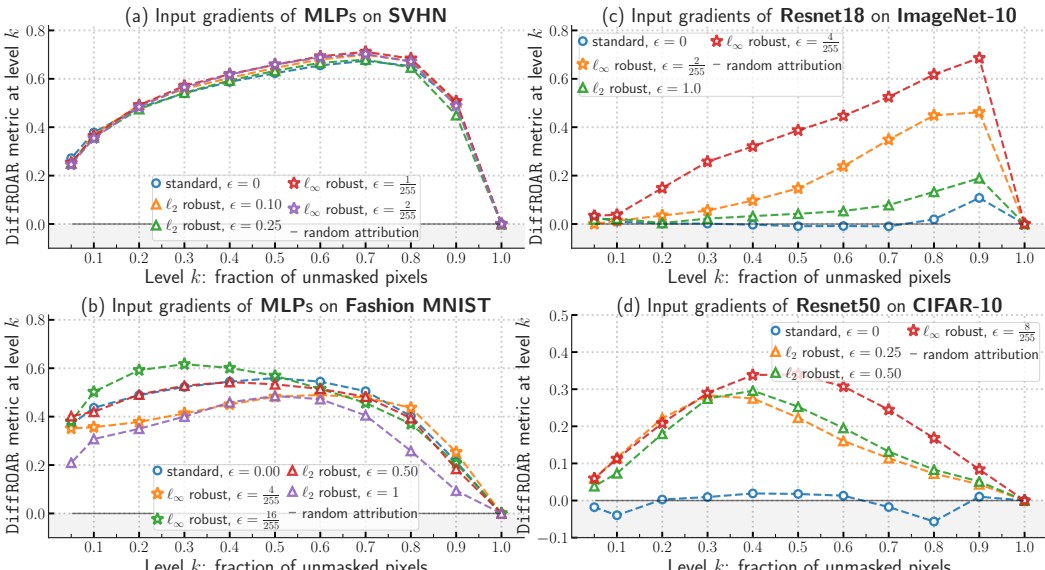

Figure 12: `DiffROAR` **results without retraining**. While we observe that standard models violate (**A**) while adversarially trained models satisfy (**A**) for the Resnet models, we see that both standard and adversarially trained models satisfy (**A**) for MLP models, showing that this evaluation methodology does not yield consistent results across model architectures/datasets. Further, the `DiffROAR` metric may be unrealiable for small unmasking fractions since this incurs heavy distribution shift. Consequently, we employ `DiffROAR` after retraining on the new unmasked data.

## C.8 Imagenet-10 images unmasked using input gradients attributions of Resnet18 models

Recall that in Section 4, we showed that unlike input gradients of standard models, robust models consistently satisfy assumption (**A**). That is, input gradients of robust models highlight discriminative features, whereas input gradients of standard models tend to highlight non-discriminative features and suppress discriminative task-relevant features. In this section, we qualitatively substantiate these findings by visualizing ImageNet-10 images that are unmasked using top-$k$ and bottom-$k$ input gradient attributions of standard and robust Resnet18 models. *Please note that the following visual assessments are only meant to* qualitatively *support findings made in Section 4 using the evaluation framework described in Section 3.* As discussed in Section 3, if input gradients attain high-magnitude `DiffROAR` score, images unmasked using top-$k$ attributions should highlight discriminative features, whereas images unmasked using bottom-$k$ should highlight non-discriminative features.

We make two observations using Figure 13 that qualitatively support our empirical findings in Section 4. First, we observe that images unmasked using top-$k$ gradient attributions of robust models tend to highlight salient aspects of images (e.g., shape of fruit or face of monkey in Figure 13), whereas bottom-$k$ attributions often mask salient aspects of images either completely or partially. Second, images unmasked using top-$k$ and bottom-$k$ attributions using input gradients of standard models exhibit visual commonalities, supporting the fact that for standard models, `DiffROAR` is close to $0$ for multiple values of $k$.

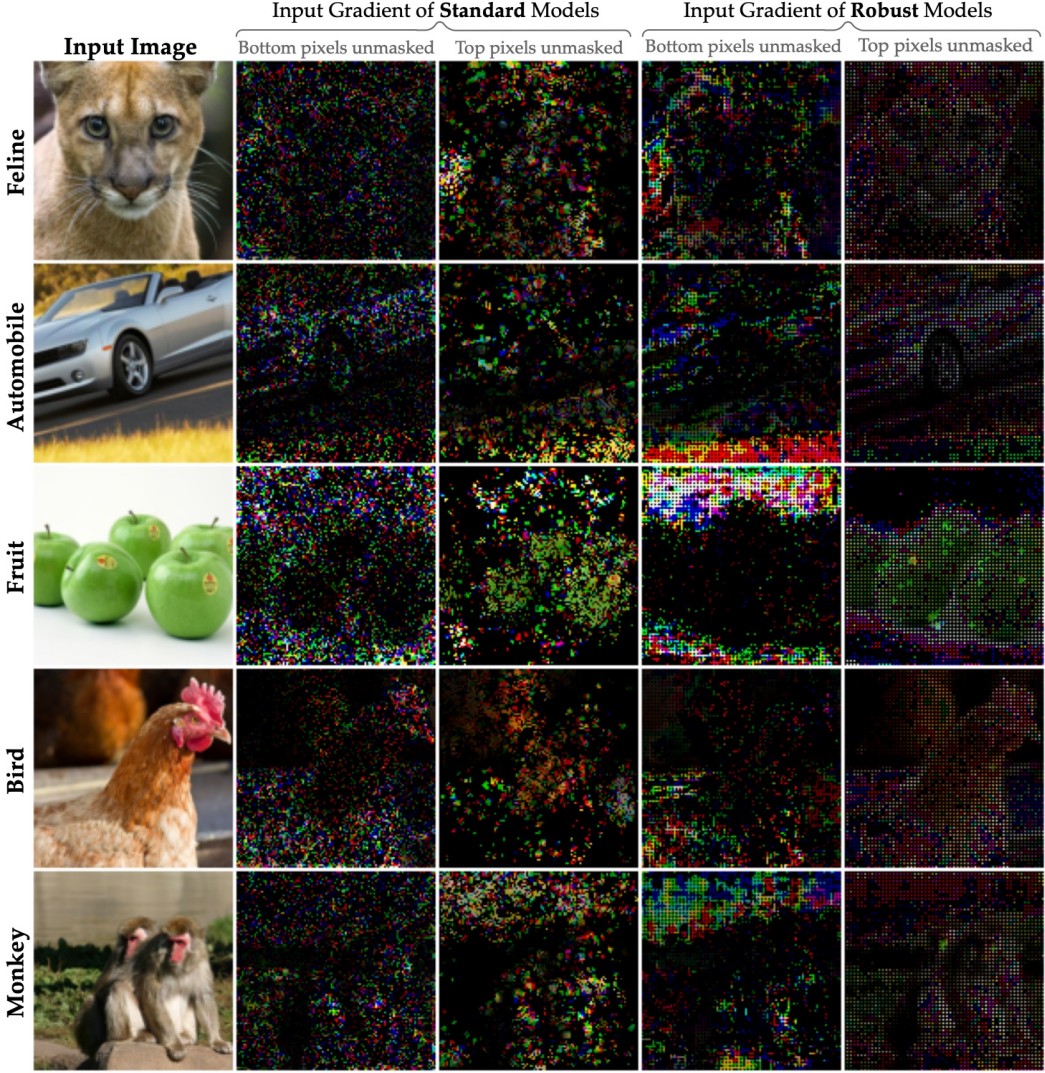

Figure 13: **ImageNet10 images unmasked using input gradient attributions**. Visualizing ImageNet-10 images that are unmasked using unmasking fraction, or level, $k = 15\%$ using input gradient attributions of standard and $\ell_\infty$-robust Resnet18 models. Top-$k$ unmasked images (i.e., images in which *only* "top" gradient attributions are unmasked) and bottom-$k$ unmasked images, attained via input gradients of standard models, share visual commonalities, suggestive of poor attribution quality. Unlike bottom-$k$ unmasked images, images unmasked using top-$k$ attributions of robust models' input gradients highlight salient aspects of images. See Appendix C.8 for details.

# D Additional experiments on feature leakage and `BlockMNIST` data

In this section, we first provide additional evidence that supports the feature leakage hypothesis in the setting used in Section 5: `BlockMNIST` data with `MNIST` digits 0 and 1 corresponding to the signal block in class 0 and class 1 respectively. Then, we show that our results vis-a-vis feature leakage and `BlockMNIST` are robust to the choice of `MNIST` digits used in the signal block as well as the number of classes in the `BlockMNIST` classification task. Finally, we end with a brief description of experiments that we conducted in order to test another hypothesized cause to understand why input gradients of standard models tend to violate (`A`).

## D.1 Additional analysis to demonstrate feature leakage in `BlockMNIST` data

In this section, we provide (i) additional examples of `BlockMNIST` images and inputs gradients of standard and robust models, (ii) additional examples of `BlockMNIST-Top` images and input gradients, and (iii) describe a *proxy* metric to measure feature leakage in `BlockMNIST`-based data.

Figure 14 shows 40 `BlockMNIST` images in the first row and their corresponding input gradients for standard and robust MLPs and Resnet18 models in the subsequent rows. We observe that input gradient attributions of standard MLP and Resnet18 models consistently highlight the signal block *as well as* the non-discriminative null block for all images. On the other hand, input gradient attributions of $\ell_2$ robust MLP and Resnet18 models exclusively highlight `MNIST` digits in the signal block and clearly suppress the square patch in the null block. These results further substantiate our results in Figure 3 by showing that unlike standard models, adversarially robust models satisfy (`A`) on `BlockMNIST` data. Figure 17 provides 20 `BlockMNIST-Top` images in the first row and the corresponding input gradients of standard MLP and Resnet models in the subsequent rows. As shown in Figure 16, in this setting, in contrast to results on `BlockMNIST`, input gradients of *standard* Resnet18 and MLP models trained on `BlockMNIST-Top` satisfy assumption (`A`).

We further substantiate these findings using a *proxy* metric to quantitatively measure feature leakage in `BlockMNIST`-based datasets. As discussed in Section 5, in the `BlockMNIST` setting, we can restate assumption (`A`) as follows: *Do input gradient attributions highlight the signal block over the null block?* We measure the extent to which input gradients of a given trained model satisfies assumption (`A`) by evaluating the fraction of top-$k$ attributions that are placed in the null block. In Figure 16, we show that the fraction of top-$k$ attributions in the null block, when averaged over all images in the test dataset, is significantly greater for standard MLPs & CNNs than for robust MLP & CNNs. In Figure 17, we show that input gradient attributions of standard models trained on `BlockMNIST-Top` place significantly fewer attributions in the null block, compared to attributions of standard models trained on `BlockMNIST`. In both cases, the proxy metric further validates our findings vis-a-vis input gradients of standard & robust models and feature leakage.

## D.2 Effect of choice and number of classes in `BlockMNIST` data

In this section, we show that our analysis on `BlockMNIST`-based datasets in Section 5 is robust to the choice and number of classes in `BlockMNIST` data. In particular, we reproduce our empirical findings vis-a-vis feature leakage and input gradient attributions of standard vs. robust models on three additional `BlockMNIST`-based tasks. In Figure 18 and Figure 19, we evaluate input gradients of standard and robust models trained on `BlockMNIST` and `BlockMNIST-Top` data, wherein the `MNIST` digits in class 0 and class 1 correspond to digits 2 and 4 (in the signal block) respectively. Similarly, in Figure 20 and Figure 21, we reproduce our empirical findings from Section 5 on `BlockMNIST` and `BlockMNIST-Top` data in which the `MNIST` digits in class 0 and class 1 correspond to digits 3 and 7 (in the signal block) respectively. In Figure 22 and Figure 23, we show that (i) input gradients of standard models violate assumption (`A`) due to feature leakage and (ii) adversarial training mitigates feature leakage on 10-class `BlockMNIST` and `BlockMNIST-Top` data, wherein each class $i\{0, \ldots, 9\}$ corresponds to `MNIST` digit $i$ in the signal block.

## D.3 Does randomness in initialization explain why input gradients violate (`A`)?

In this section, *we investigate whether the poor quality of input gradients in standard models is due to randomness retained from the initialization*. Figure 24 shows scatter plots of input gradient values over all pixels in all images before (x-axis) and after (y-axis) standard training on four image

classification benchmarks. The results indicate that (i) the scale of gradients after training is at least an order of magnitude larger than those before training and (ii) the gradient values before and after training are uncorrelated. Together, these results suggest that random initialization does not have much of a role in determining the input gradients after training.

**D.4  Do other feature attribution methods exhibit feature leakage?**

In this section, we evaluate feature leakage in five feature attribution methods: Integrated Gradients [22], Layer-wise Relevance Propagation (LRP) [24], Guided Backprop [16], Smoothgrad [2] (with standard deviation $\sigma \in \{0.1, 0.3, 0.5\}$), and Occlusion [67] (with patch size $\rho \in \{5, 10\}$). First, we evaluate the aforementioned feature attribution methods on standard models trained on `BlockMNIST` data. As shown in Figure 25 and Figure 26, in addition to vanilla input gradients, all five feature attribution methods evaluated on standard MLPs and Resnet18 models highlight the `MNIST` signal block as well as the null block. Conversely, Figure 27 and Figure 28 show that when standard MLPs and Resnet18 models are trained on `BlockMNIST-Top` data, all feature attribution methods exclusively highlight the `MNIST` signal block. These results collectively indicate that similar to vanilla input gradient attributions, multiple feature attribution methods exhibit feature leakage. Furthermore, consistent with our findings on adversarial robustness vis-a-vis feature leakage, Figure 29 and Figure 30 show that feature attribution method evaluated on adversarially robust MLPs and Resnet18 model do not exhibit feature leakage on `BlockMNIST` data.

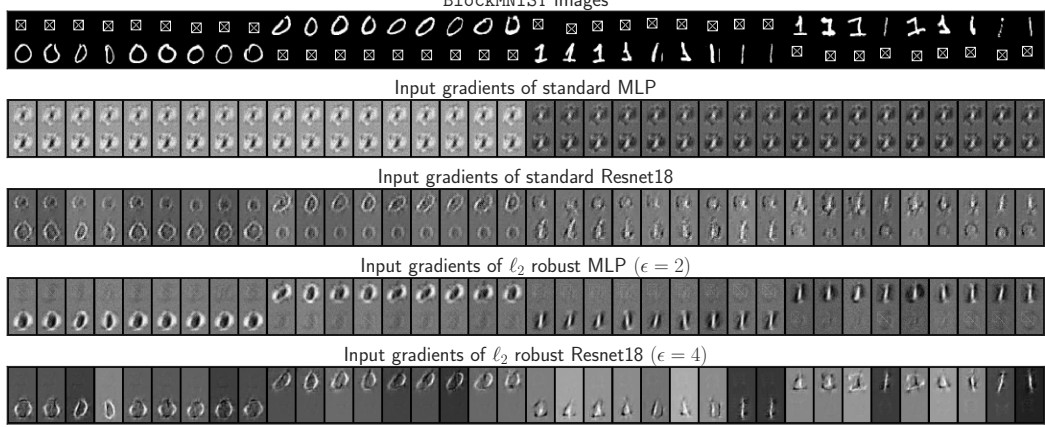

Figure 14: **BlockMNIST 0 vs. 1**. 40 `BlockMNIST` (MNIST $0$ vs. $1$) images and their corresponding input gradients. Recall that every image consists of a *signal* and *null* block, each randomly placed at the *top* or *bottom*. The *signal* block, containing the `MNIST` digit $0$ or $1$, determines the image class, $0$ or $1$. The *null* block, containing the square patch, does not encode any information of the image class. The second, third, and fourth rows show input gradients of standard Resnet18, standard MLP, $\ell_2$ robust Resnet18 ($\epsilon = 2$) and $\ell_2$ robust MLP ($\epsilon = 4$) respectively. The plots clearly show that input gradients of standard `BlockMNIST` models incorrectly highlight *the non-discriminative null block* as well, thereby violating (A). In contrast, input gradients of robust models highlight the signal block, suppress the null block, and satisfy (A). See Appendix D.1 for details.

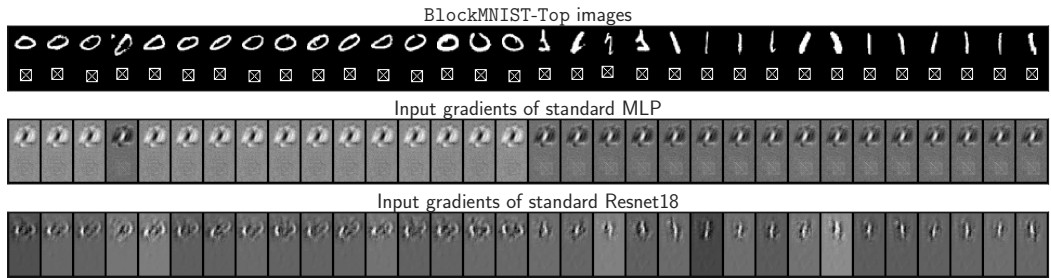

Figure 15: **BlockMNIST-Top 0 vs. 1**. 20 `BlockMNIST-Top` (MNIST $0$ vs. $1$) images and input gradients of standard MLP and Resnet18 models. As shown in the first row, the signal & null blocks are fixed at the top & bottom respectively in `BlockMNIST-Top` images. In contrast to results on `BlockMNIST` in fig. 1, input gradients of standard models trained on `BlockMNIST-Top` highlight the signal block, suppress the null block, and satisfy (A). Please see Appendix D.1 for details.

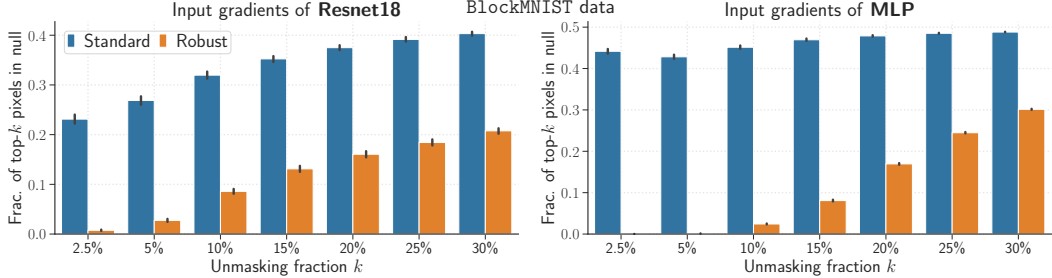

Figure 16: **Proxy metric to compare input gradients of standard and robust models trained on** `BlockMNIST` **($0$ vs. $1$) data**. The proxy metric measures the fraction of top-$k$ attributions that are placed in the null block of images in the test dataset. The left and right subplots evaluate this metric on input gradient attributions of standard and robust Resnet18 models and MLPs respectively. Compared to input gradients of standard models, adversarially trained models place significantly fewer top-$k$ attributions in the null block for multiple values of unmasking fraction $k$. Details in Appendix D.1.

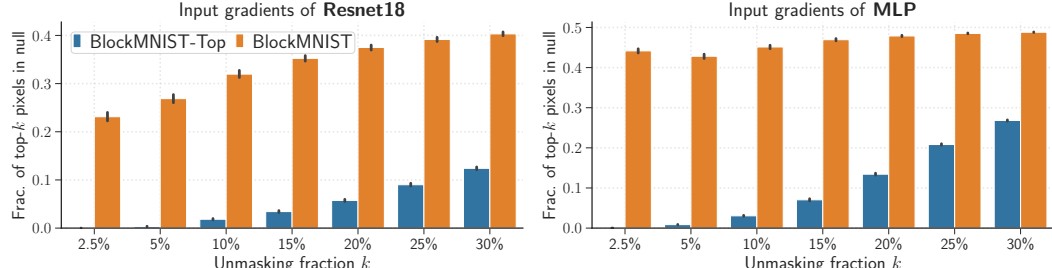

Figure 17: **Proxy metric to compare input gradients of standard models trained on** `BlockMNIST` **and** `BlockMNIST-Top` **(0 vs. 1) data**. The proxy metric measures the fraction of top-$k$ attributions that are placed in the null block of images. The left and right subplots evaluate this metric on input gradient attributions of standard Resnet18 models and MLPs trained on `BlockMNIST` and `BlockMNIST-Top` data respectively. Compared to input gradients of models trained on `BlockMNIST`, standard models trained on `BlockMNIST-Top` place significantly fewer top-$k$ attributions in the null block for multiple values of unmasking fraction $k$. Details in Appendix D.1.

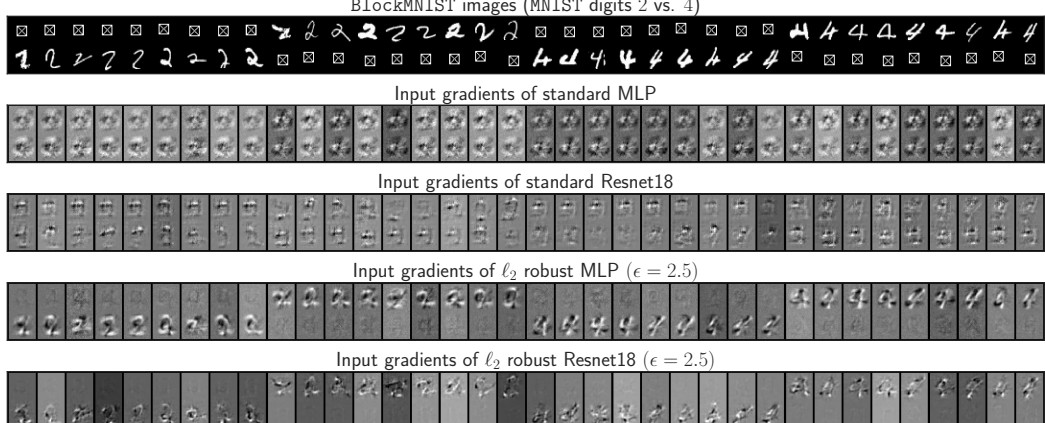

Figure 18: **BlockMNIST 2 vs. 4**. 40 `BlockMNIST` (MNIST $2$ vs. $4$) images and their corresponding input gradients. The *signal* block, containing the MNIST digit 2 or 4, determines the image class, 0 or 1. The second, third, and fourth rows show input gradients of standard Resnet18, standard MLP, $\ell_2$ robust Resnet18 ($\epsilon = 2.5$) and $\ell_2$ robust MLP ($\epsilon = 2.5$) respectively. The plots clearly show that input gradients of standard `BlockMNIST` models incorrectly highlight *the non-discriminative null block* as well, thereby violating (**A**). In contrast, input gradients of robust models highlight the signal block, suppress the null block, and satisfy (**A**). See Appendix D.1 for details.

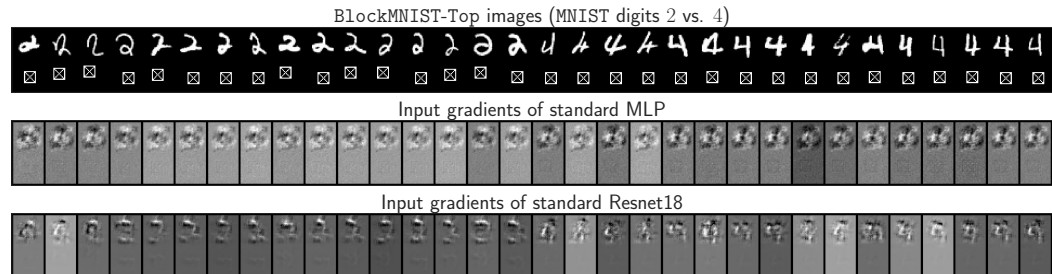

Figure 19: **BlockMNIST-Top 2 vs. 4**. 20 `BlockMNIST-Top` (MNIST $2$ vs. $4$) images and corresponding input gradients of standard MLP and Resnet18 models. As shown in the first row, the signal & null blocks are fixed at the top & bottom respectively in `BlockMNIST-Top` images. In contrast to results on `BlockMNIST` in fig. 1, input gradients of standard models trained on `BlockMNIST-Top` highlight the signal block, suppress the null block, and satisfy (**A**). Please see Appendix D.1 for details.

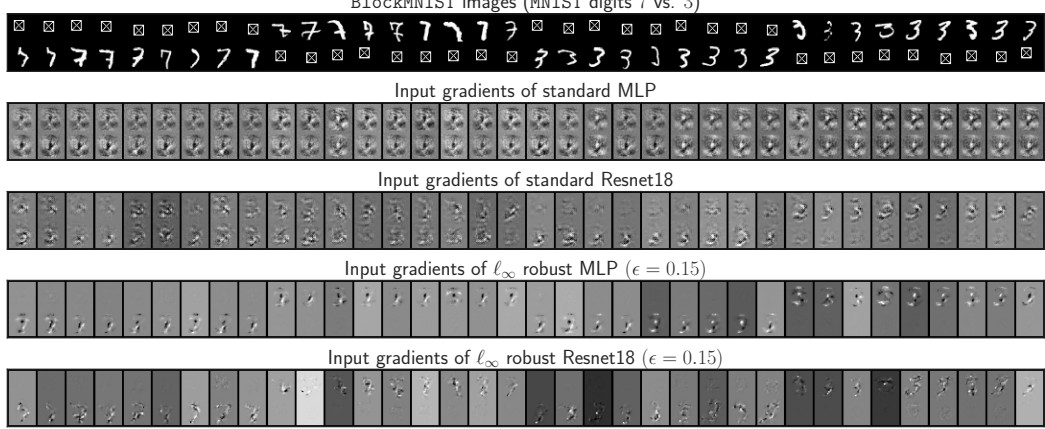

Figure 20: **BlockMNIST 3 vs. 7**. 40 `BlockMNIST` (MNIST 3 vs. 7) images and their corresponding input gradients. The *signal* block, containing the `MNIST` digit 3 or 7, determines the image class, 0 or 1. The second, third, and fourth rows show input gradients of standard Resnet18, standard MLP, $\ell_\infty$ robust Resnet18 ($\epsilon = 0.15$) and $\ell_\infty$ robust MLP ($\epsilon = 0.15$) respectively. The plots clearly show that input gradients of standard `BlockMNIST` models incorrectly highlight *the non-discriminative null block* as well, thereby violating (**A**). In contrast, input gradients of robust models highlight the signal block, suppress the null block, and satisfy (**A**). See Appendix D.1 for details.

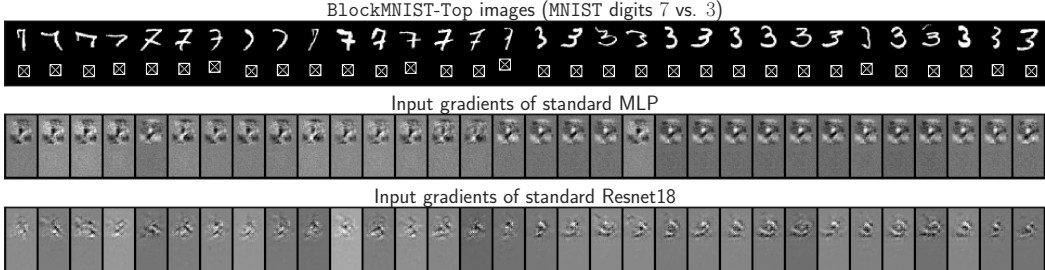

Figure 21: **BlockMNIST-Top 3 vs. 7**. 20 `BlockMNIST-Top` (MNIST 3 vs. 7) images and input gradients of standard MLP and Resnet18 models. As shown in the first row, the signal & null blocks are fixed at the top & bottom respectively in `BlockMNIST-Top` images. In contrast to results on `BlockMNIST` in fig. 1, input gradients of standard models trained on `BlockMNIST-Top` highlight the signal block, suppress the null block, and satisfy (**A**). Please see Appendix D.1 for details.

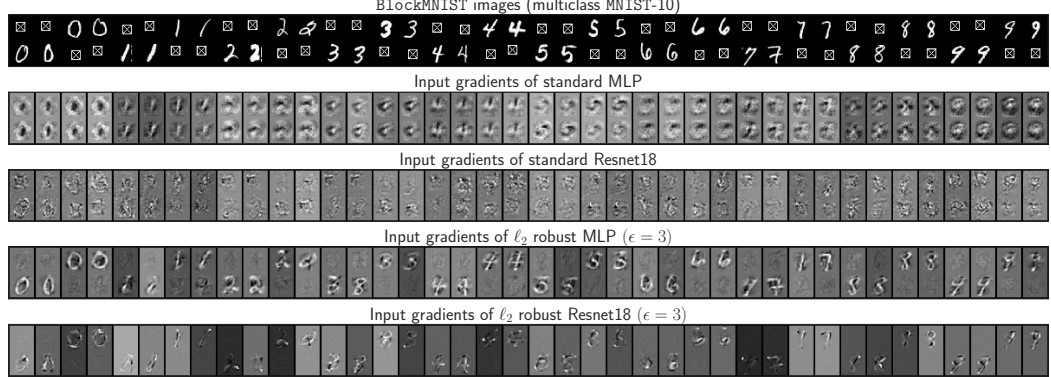

Figure 22: **Multiclass BlockMNIST**. 40 `BlockMNIST` (all `MNIST` classes) images and their corresponding input gradients. dataset. In this setting, the *signal* block, containing an `MNIST` digit sampled from a class chosen uniformly at random, determines the image class $y \in \{0, \ldots, 9\}$. The second, third, and fourth rows show input gradients of standard Resnet18, standard MLP, $\ell_2$ robust Resnet18 ($\epsilon = 3$) and $\ell_2$ robust MLP ($\epsilon = 3$) respectively. The plots clearly show that input gradients of standard `BlockMNIST` models incorrectly highlight *the non-discriminative null block* as well, thereby violating (A). In contrast, input gradients of robust models highlight the signal block, suppress the null block, and satisfy (A). See Appendix D.1 for details.

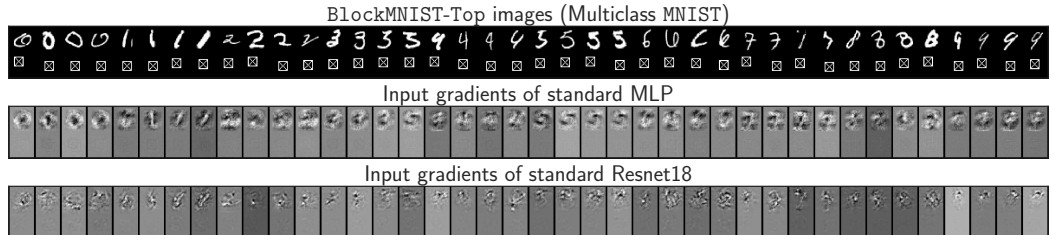

Figure 23: **Multiclass BlockMNIST-Top**. 40 `BlockMNIST-Top` (all `MNIST` classes) images and corresponding input gradients of standard MLP and Resnet18 models. As shown in the first row, the signal & null blocks are fixed at the top & bottom respectively in `BlockMNIST-Top` images. In contrast to results on `BlockMNIST` in fig. 1, input gradients of standard models trained on `BlockMNIST-Top` highlight the signal block, suppress the null block, and satisfy (A). Please see Appendix D.1 for details.

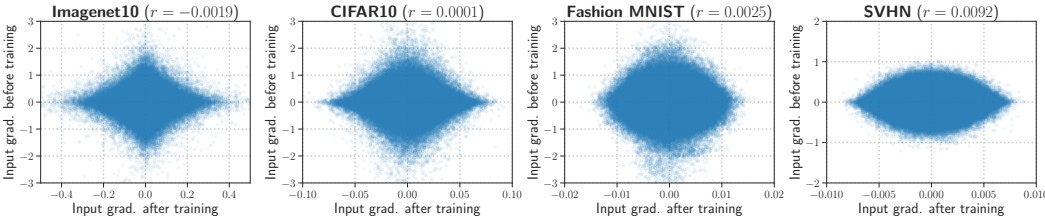

Figure 24: **Does random initialization affect input gradients after training?** The scatter plots above show the gradient values at the beginning of training and after end of training on y-axis and x-axis respectively. We can see that the scale of gradients is much larger at the end of training compared to that at the beginning of training and both of them are uncorrelated. This suggests that the poor quality of input gradients of standard trained models is a result of the training process, and not because of random initialization.

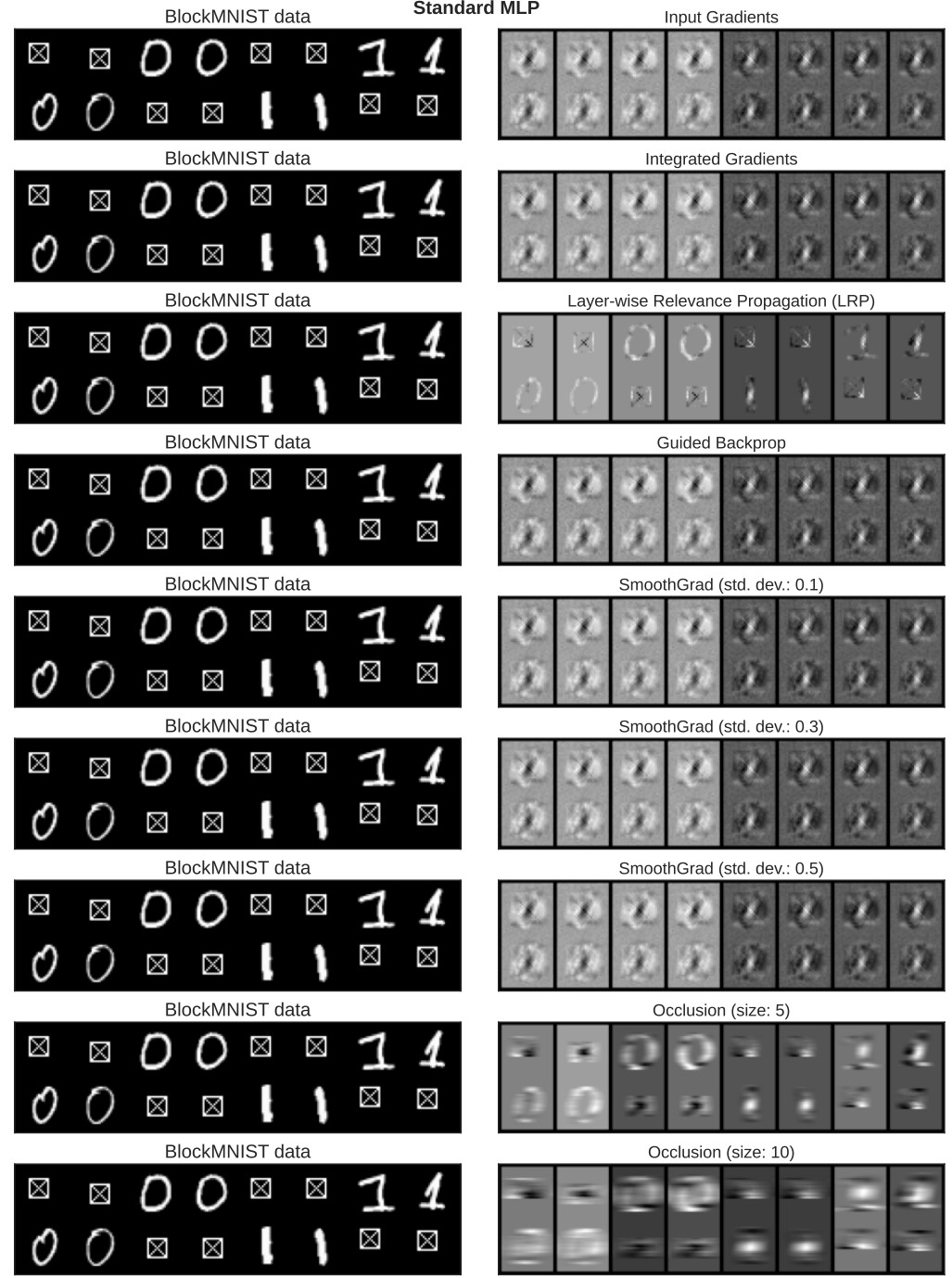

Figure 25: Multiple instance-specific feature attribution methods evaluated using a standard two-layer MLP trained on BlockMNIST data. All feature attribution methods exhibit feature leakage, as the attributions highlight the non-predictive null block in addition to the MNIST signal block.

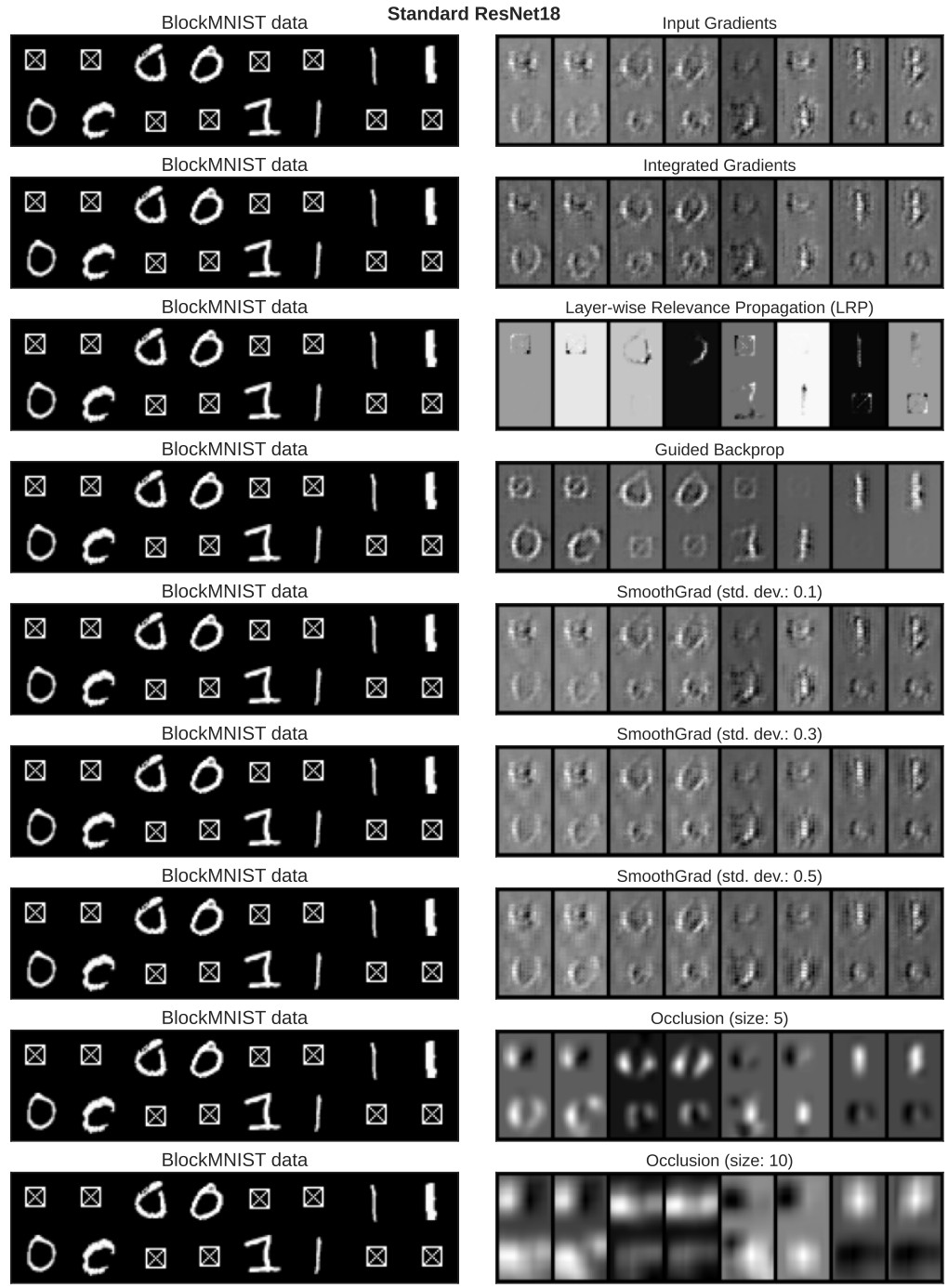

Figure 26: Multiple instance-specific feature attribution methods evaluated using a standard ResNet18 trained on `BlockMNIST` data. All feature attribution methods exhibit feature leakage, as the attributions highlight the non-predictive null block in addition to the `MNIST` signal block. Surprisingly, in some cases, LRP (third row) exclusively highlights the null block.

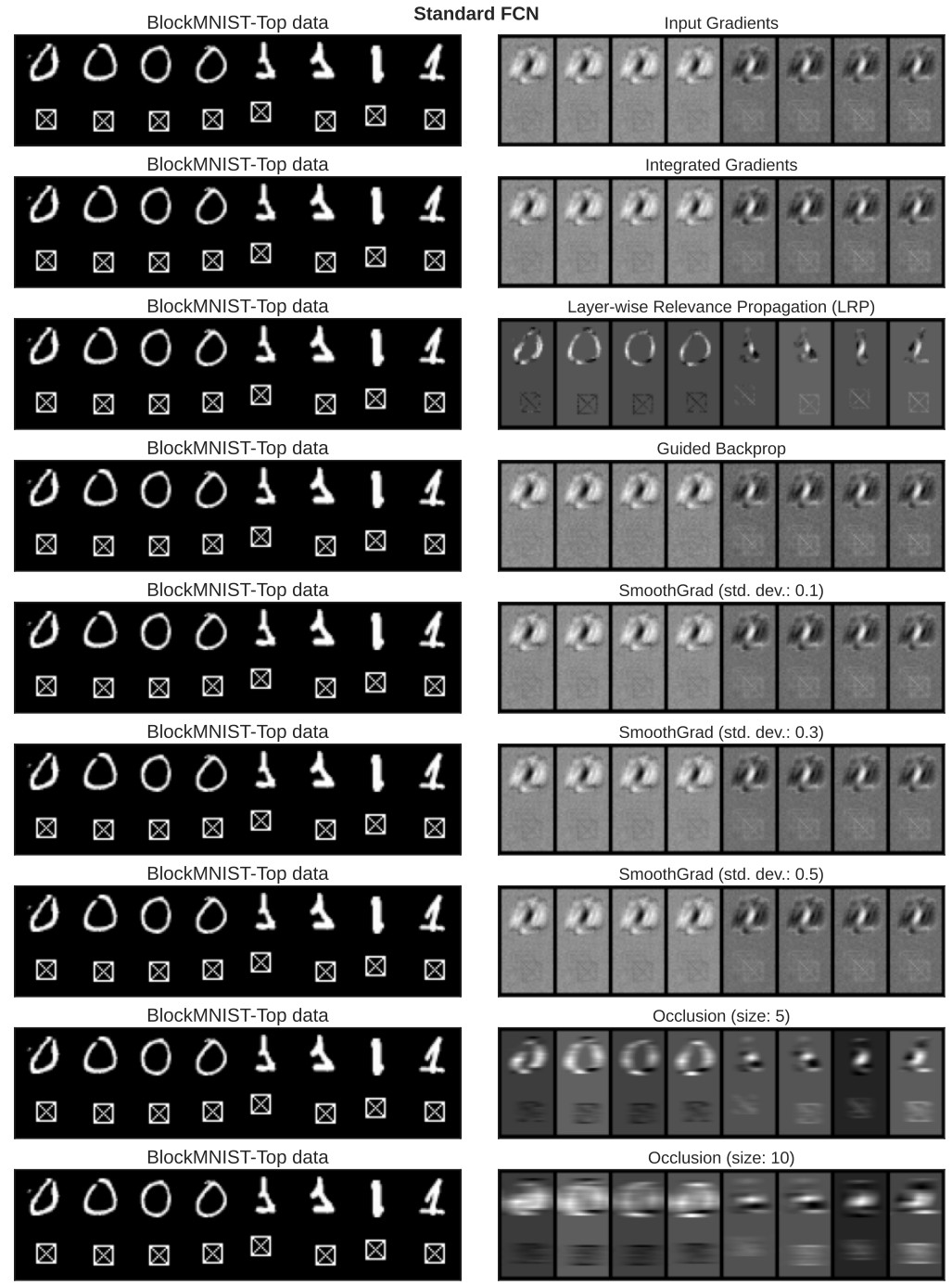

Figure 27: Multiple instance-specific feature attribution methods evaluated using a standard two-layer MLP trained on `BlockMNIST-Top` images, in which the `MNIST` signal block is fixed at the top. On this dataset, feature attributions of all five methods highlight discriminative features in the signal block, suppress the null block, and satisfy (`A`).

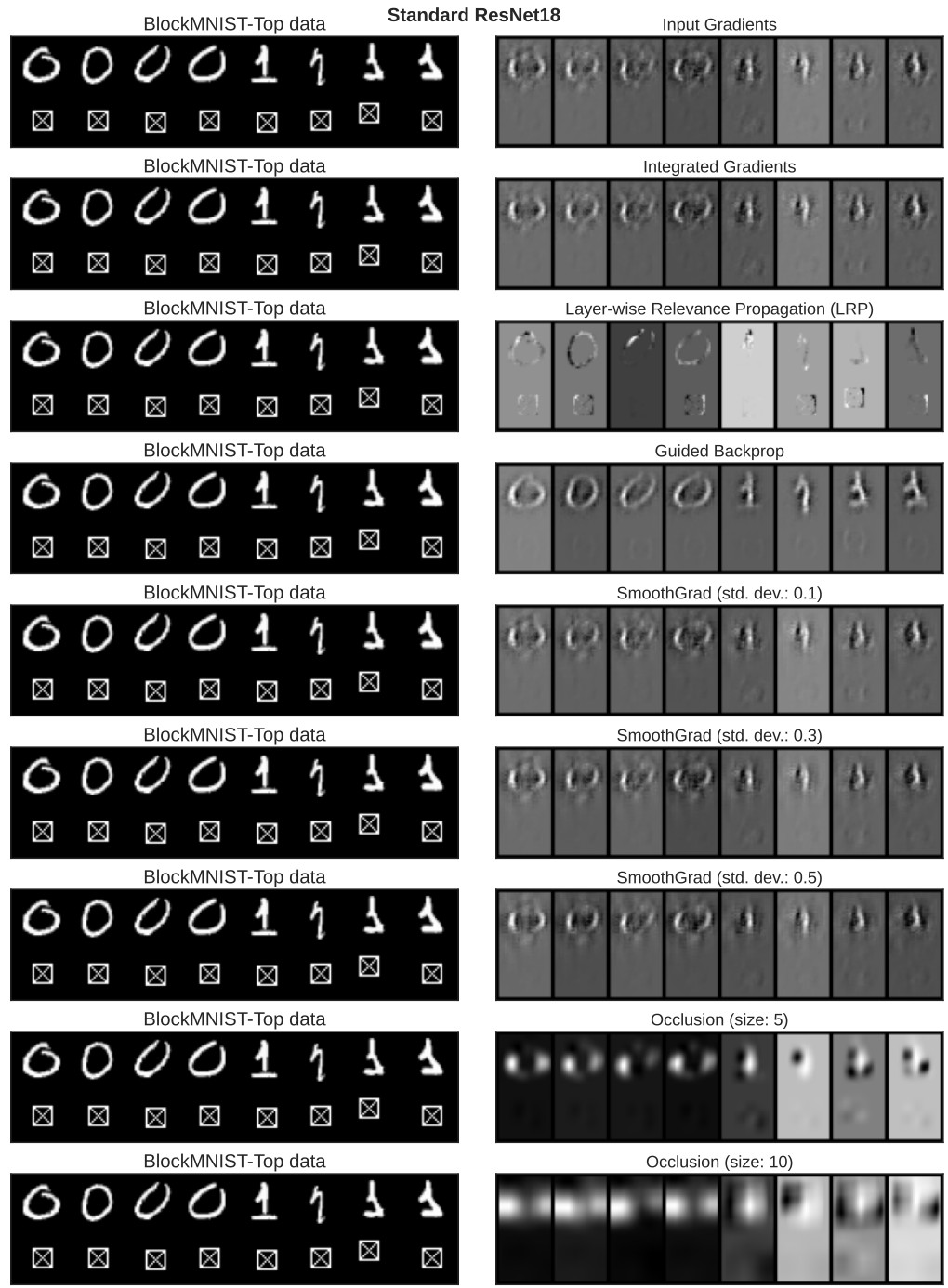

Figure 28: Multiple instance-specific feature attribution methods evaluated using a standard ResNet18 trained on `BlockMNIST-Top` data, in which the `MNIST` signal block is fixed at the top. On this dataset, feature attributions of all five methods highlight discriminative features in the signal block, suppress the null block, and satisfy (`A`). Surprisingly, `LRP` attributions (third row) highlight the null patch of `BlockMNIST-Top` images as well.

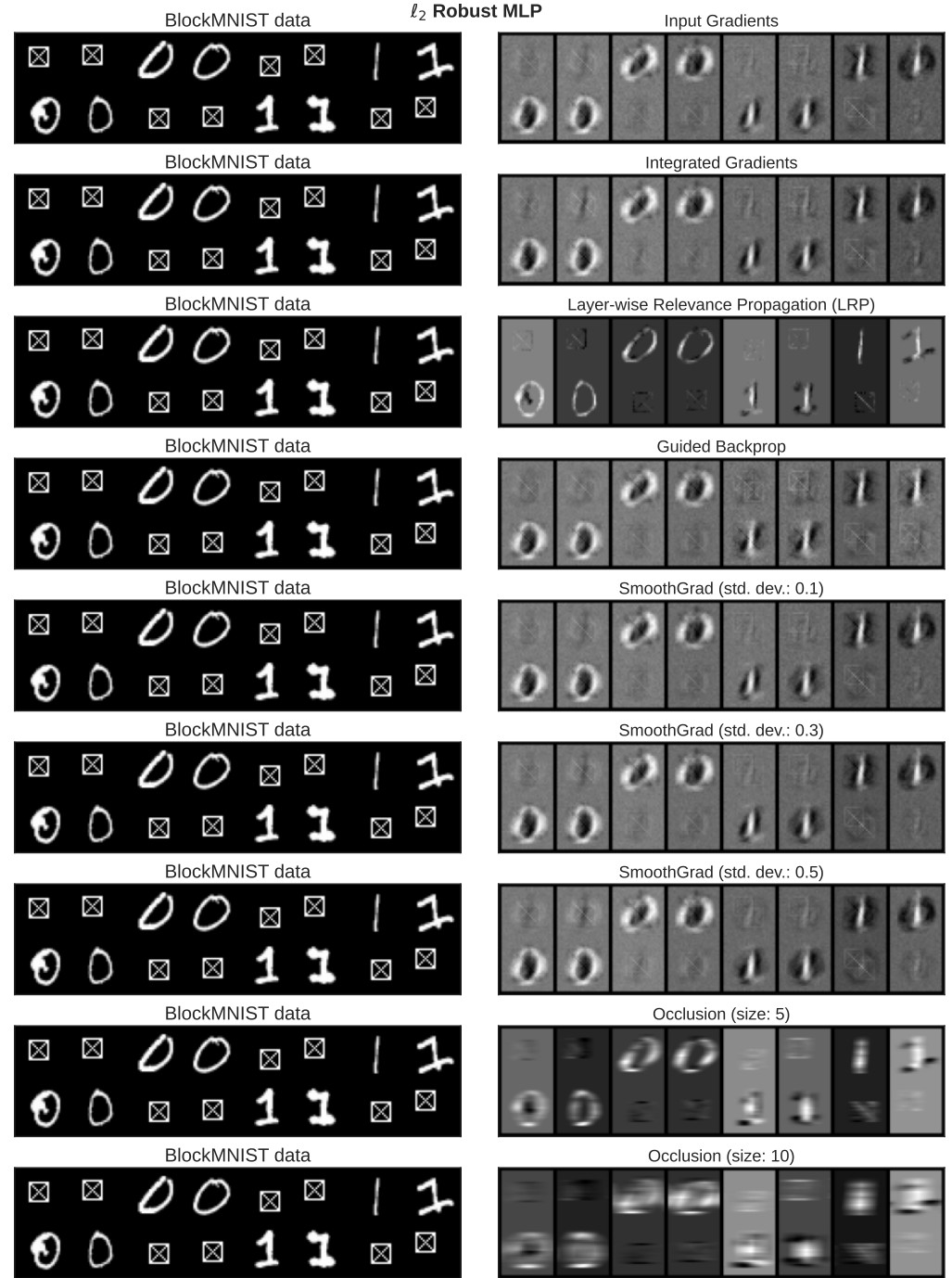

Figure 29: Multiple instance-specific feature attribution methods evaluated using a $\ell_2$ robust two-layer MLP trained on `BlockMNIST` data. Consistent with our findings on adversarial robustness vis-a-vis feature leakage, feature attributions of all methods of robust models do not exhibit feature leakage.

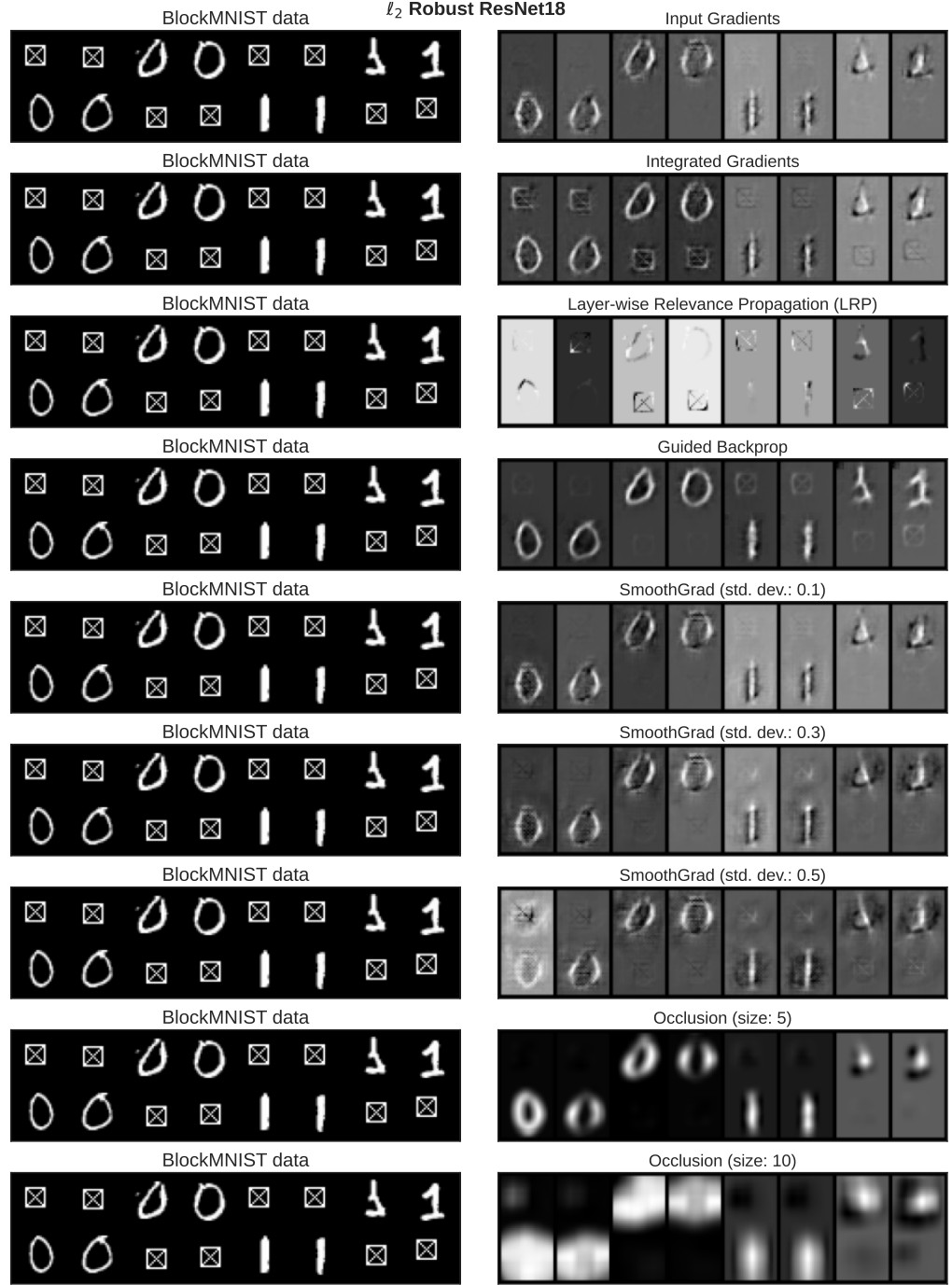

Figure 30: Multiple instance-specific feature attribution methods evaluated using a $\ell_2$ robust ResNet18 trained on BlockMNIST data. Consistent with our findings on adversarial robustness vis-a-vis feature leakage, feature attributions of all methods (except LRP) of robust models do not exhibit feature leakage.

# E  Proof of Theorem 1

We first begin with the definition of a function, $\psi : \mathbb{R}^2 \to \mathbb{R}$ which will prove useful in the analysis:

$$\psi(a,b) := \phi(a+b) - \phi(-a+b) = \begin{cases} a - b & \text{if } a \leq -|b| \\ 0 & \text{if } b \leq 0, \ |a| \leq |b| \\ 2a & \text{if } b \geq 0, \ |a| \leq |b| \\ a + b & \text{if } a \geq |b| \end{cases}, \tag{5}$$

where we recall that $\phi(a) = \max(a,0)$ is the ReLU nonlinearity.

*Proof of Theorem 1 in the rich regime.* We first claim that the max-margin classifier (4) is given by $\nu^* = \frac{1}{2}\delta_{\theta_1^*} + \frac{1}{2}\delta_{\theta_2^*}$, where $\theta_1 := \left( \frac{1}{\sqrt{2}}, \frac{1}{\sqrt{2((d/2)+(1-(\eta d/2)^2)}} z, \frac{(1-(\eta d/2))}{\sqrt{2((d/2)+(1-(\eta d/2))^2)}} \right)$ and $\theta_2 := \left( \frac{-1}{\sqrt{2}}, \frac{-1}{\sqrt{2((d/2)+(1-(\eta d/2))^2)}} z, \frac{(1-(\eta d/2))}{\sqrt{2((d/2)+(1-(\eta d/2))^2)}} \right)$ with $z \in \mathbb{R}^{\widetilde{d} \cdot d}$ denoting the concatenation of $d/2$ copies of $u^* \in \mathbb{R}^{\widetilde{d}}$ and $d/2$ copies of $0$ vectors of dimension $\widetilde{d}$ each. To do so, we use [48, Proposition 12] which requires us to verify that there exists a probability distribution $p^*$ over the training data points such that:

$$\text{Support}(\nu^*) \in \underset{(w,r,b) \in \mathbb{S}^{d\widetilde{d}+1}}{\arg\max} \ \mathbb{E}_{(x,y) \sim p^*} \left[ y \cdot (w\phi(\langle r, x \rangle + b)) \right], \text{ and} \tag{6}$$

$$\text{Support}(p^*) \in \underset{(x,y) \in \mathcal{D}}{\arg\min} \ y \cdot \mathbb{E}_{(w,r,b) \sim \nu^*} \left[ w\phi(\langle r, x \rangle + b) \right], \tag{7}$$

where $\mathbb{S}^{d\widetilde{d}+1}$ denotes the unit sphere in $\mathbb{R}^{d\widetilde{d}+2}$. In order to verify this condition, we use $p^* = \frac{1}{d} \sum_{\substack{j \in [d/2] \\ y \in \{\pm 1\}}} \delta_{(y\widetilde{u}_j, y)}$, where $\widetilde{u}_j \in \mathbb{R}^{\widetilde{d} \cdot d}$ is defined as the concatenation of $d$ vectors, each in $\mathbb{R}^{\widetilde{d}}$, with the $j^{\text{th}}$ one being $u^*$, the remaining $[d/2] \setminus \{j\}$ being $-\eta u^*$ and the last $[d/2]$ being all zero vectors. We first prove (7). Consider the point $(\widehat{u}, y = 1)$ in the support of the training distribution with $\widehat{u} = (u^* + \eta g_1, \eta g_2, \cdots, \eta g_{d/2}, 0, \cdots, 0)$. We see that:

$$y \cdot \mathbb{E}_{(w,r,b) \sim \nu^*} \left[ w\phi(\langle r, \widehat{u} \rangle + b) \right]$$

$$= \frac{1}{2} \cdot \frac{1}{\sqrt{2}} \left( \phi \left( \frac{\langle z, \widehat{u} \rangle}{\sqrt{2((d/2) + (1 - (\eta d/2))^2)}} + \frac{1 - (\eta d/2)}{\sqrt{2((d/2) + (1 - (\eta d/2))^2)}} \right) \right.$$

$$\left. - \phi \left( \frac{-\langle z, \widehat{u} \rangle}{\sqrt{2((d/2) + (1 - (\eta d/2))^2)}} + \frac{1 - (\eta d/2)}{\sqrt{2((d/2) + (1 - (\eta d/2))^2)}} \right) \right)$$

Since $\langle z, \widehat{u} \rangle = 1 + \eta \sum_{i \in [d/2]} \langle g_i, u^* \rangle \geq 1 - (\eta d/2) > 0$. Consequently, using (5), we have that:

$$y \cdot \mathbb{E}_{(w,r,b) \sim \nu^*} \left[ w\phi(\langle r, \widehat{u} \rangle + b) \right] \geq \frac{1}{2\sqrt{2}} \cdot \frac{2(1 - (\eta d/2))}{\sqrt{2((d/2) + (1 - (\eta d/2))^2)}} = y \cdot \mathbb{E}_{(w,r,b) \sim \nu^*} \left[ w\phi(\langle r, y\widetilde{u}_j \rangle + b) \right].$$

This proves (7).

We now prove (6). For $\theta = (w, r, b)$, denote $\mathcal{L}(\theta) := \mathbb{E}_{\mathcal{D}} \left[ y \cdot (w\phi(\langle r, x \rangle + b)) \right]$. We have $\mathcal{L}(\theta_1) = \mathcal{L}(\theta_2) = \frac{1}{d} \cdot \sum_{j \in [d/2]} \frac{1}{\sqrt{2}} \frac{2(1 - (\eta d/2))}{\sqrt{2((d/2) + (1 - (\eta d/2))^2)}} = \frac{1 - (\eta d/2)}{2\sqrt{(d/2) + (1 - (\eta d/2))^2}}$. We will now show that $\max_{\theta \in \mathbb{S}^{d+1}} \mathcal{L}(\theta) = \frac{1 - (\eta d/2)}{2\sqrt{(d/2) + (1 - (\eta d/2))^2}}$. For a given $\theta = (w, r, b)$, we first show that it is sufficient to consider the case where $r$ is the concatenation of $\alpha_i u^*$ for some $\alpha_1, \cdots, \alpha_d$. In order to do this, given any $\theta = (w, r, b)$, let $\alpha_i := \langle r_i, u^* \rangle$ for $i \in [d/2]$ denote the inner product of the $i^{\text{th}}$-block vector of $r$ with $u^*$ and let $\bar{\alpha} := \frac{1}{d/2} \sum_{i \in [d/2]} \alpha_i$ be the mean of $\alpha_i$. The function $\mathcal{L}(\theta)$ can be simplified as:

$$\mathcal{L}(\theta) = \frac{w}{d} \sum_{i \in [d/2]} \left( \phi(\alpha_i - (\eta d/2)\bar{\alpha} + b) - \phi(-\alpha_i + (\eta d/2)\bar{\alpha} + b) \right).$$

We can now consider $r'$ to be the concatenation of $\langle r_i, u^* \rangle u^*$ for $i \in [d/2]$ and the remaining coordinates equal to zero, which ensures that $\mathcal{L}(\theta) = \mathcal{L}(\theta')$ for $\theta' = (w, r', b)$ and $\|\theta\| \geq \|\theta'\|$. We

can then choose $|w'| \geq |w|$ such that $\|(w', r', b)\| = 1$ and $\mathcal{L}((w', r', b)) > \mathcal{L}(\theta)$. So it suffices to consider $\mathcal{L}(\theta)$ for $\theta = (w, r, b)$ where $r$ is the concatenation of $\alpha_i u^*$ for some $\alpha_i$ for $i \in [d/2]$ and the remaining coordinates being set to zero. Let us consider two situations separately:

**Case I,** $b \geq 0$: Recall from (5) the definition $\psi(a, b) := \phi(a + b) - \phi(-a + b)$. First note from (5) that, $|\psi(a, b) - \psi(a', b)| \leq 2|a - a'|$ and $\psi(a, b) - \psi(a', b) \geq a - a'$ for every $a > a'$. If $\alpha_j < 0$ for some $j$, then choosing $r'_j = r_j - 2\alpha_i w^*$ with $r'_i = r_i$ for all $i \neq j$ gives us a corresponding $\theta'$ satisfying

$$
\begin{aligned}
\mathcal{L}(\theta') &\geq \frac{w}{d} \sum_{i \neq j} \left( \phi\left(\alpha_i - (\eta d/2)\bar{\alpha} + b\right) - \phi\left(-\alpha_i + (\eta d/2)\bar{\alpha} + b\right) \right) - 2\eta(d/2 - 1)|\alpha_i| \\
&\quad + \frac{w}{d} \left( \phi\left(\alpha_i - (\eta d/2)\bar{\alpha} + b\right) - \phi\left(-\alpha_i + (\eta d/2)\bar{\alpha} + b\right) \right) + (1 - \eta)|\alpha_i| \\
&\geq \mathcal{L}(\theta) + \frac{w}{d} \cdot (1 - \eta d) \cdot |\alpha_i|.
\end{aligned}
$$

So, it suffices to restrict our attention to $\theta$ such that $\alpha_i \geq 0$ for all $i$ in order to prove (6). We will now show that making all $\alpha_i$ equal will further increase the value of $\mathcal{L}$. In order to see this, let $\alpha_1 = \min_{i \in [d/2]} \alpha_i$ and $\alpha_2 = \max_{i \in [d/2]} \alpha_i$. Then constructing $r'$ from $r$ by replacing $\alpha_1$ and $\alpha_2$ with $\alpha' := \frac{\alpha_1 + \alpha_2}{2}$ ensures that $\|r'\| \leq \|r\|$ while at the same time $\mathcal{L}((w, r', b)) - \mathcal{L}((w, r, b))$ since $\alpha_1 \geq 0$ implies $\alpha_1 - (\eta d/2)\bar{\alpha} > -(\alpha_2 - (\eta d/2)\bar{\alpha})$. If $\alpha_i = \alpha_j$ for all $i, j \in [d]$, then from (5),

$$
\mathcal{L}(\theta) = \frac{w}{d} \sum_{i \in [d]} \alpha_i - (\eta d/2)\bar{\alpha} + \min\left(\alpha_i - (\eta d/2)\bar{\alpha}, b\right) = (1 - (\eta d/2))w\left(\bar{\alpha} + \min\left(\bar{\alpha}, \frac{b}{1 - (\eta d/2)}\right)\right).
$$

The maximizer of the above expression under the constraint $\|\theta\|^2 = w^2 + (d/2)\bar{\alpha}^2 + b^2 = 1$ can be seen to be when $w = \pm\frac{1}{\sqrt{2}}$, $\bar{\alpha} = \frac{2}{\sqrt{(d/2)+(1-(\eta d/2))^2}}$ and $b = \frac{1-(\eta d/2)}{\sqrt{2((d/2)+(1-(\eta d/2))^2)}}$ achieving value $\frac{1-(\eta d/2)}{2\sqrt{(d/2)+(1-(\eta d/2))^2}}$.

**Case II,** $b < 0$: In this case, we have from (5) that

$$
\begin{aligned}
\mathcal{L}(\theta) &\leq \frac{|w|}{d} \sum_{i \in [d]} \alpha_i - (\eta d/2)\bar{\alpha} = (1 - (\eta d/2))|w|\bar{\alpha} \leq \frac{(1 - (\eta d/2))|w|\|r\|}{\sqrt{d/2}} \leq \frac{1 - (\eta d/2)}{2\sqrt{d/2}} \\
&< \frac{1 - (\eta d/2)}{\sqrt{(d/2) + (1 - (\eta d/2))^2}},
\end{aligned}
$$

where we used $\eta < \frac{1}{10d}$ in the last step. This shows that $\nu^* = \frac{1}{2}\delta_{\theta_1^*} + \frac{1}{2}\delta_{\theta_2^*}$ is a max-margin classifier satisfying (4).

**Gradient magnitude**: For any input $(x, y)$, we note that the input gradient is of the form $\nabla_x \mathcal{L}(\nu^*, (x, y)) = \alpha z$ for some $\alpha \neq 0$. Consequently, the claim about the gradient magnitudes in different coordinates follows from the structure of $z$ proved above.

$\square$

# F Effect of adversarial training

Consider training a model that is adversarially robust in an $\ell_p$ ball of radius $\epsilon$. Assuming that the inner iterations of adversarial training find the optimal perturbations, it can be shown that if adversarial training converges asymptotically (i.e., in the rich regime), it does so to an appropriate max-margin classifier [68]:

$$\nu^* := \underset{\nu \in \mathcal{P}\left(\mathbb{S}^{d\tilde{d}+1}\right)}{\arg\max} \min_{(x,y) \in B_p(\mathcal{D},\epsilon)} y \cdot f(\nu, x), \tag{8}$$

where $B_p(\mathcal{D}, \epsilon) := \left\{ (x,y) : (\tilde{x}, y) \sim \mathcal{D}, \|x - \tilde{x}\|_p \leq \epsilon \right\}$. This implies that using the techniques of previous section, we should be able to analyze the input gradient. However, such analysis requires an explicit form of the max-margin classifier defined above. In contrast to the standard training studied above, computing explicit form of the max-margin classifier is significantly non-trivial in the adversarially training case even for the simple special case of $\tilde{d} = 1, \eta = 0$ and $u^* = 1$. While we are unable to explicitly compute the max-margin classifier even for this case, we make the following conjecture about the max-margin classifier.

**Conjecture 1.** Let data distribution $\mathcal{D}$ follow (2) with $\tilde{d} = 1, \eta = 0$ and $u^* = 1$. Then, the classifier $\tilde{\nu}$ defined below is a max-margin classifier for adversarial training (8) for $p = \infty$ and $\epsilon$ close to $0.5$:

$$\tilde{\nu} := \frac{1}{d} \sum_{i \in [d/2]} \delta_{\theta_i} + \delta_{\theta_i'}, \tag{9}$$

with $\theta_i := (\frac{1}{\sqrt{2}}, \frac{3}{\sqrt{20}} e_i, \frac{-1}{\sqrt{20}}), \theta_i' := (\frac{-1}{\sqrt{2}}, \frac{-3}{\sqrt{20}} e_i, \frac{-1}{\sqrt{20}})$ where $e_i \in \mathbb{R}^d$ denotes $i^{\text{th}}$ standard basis vector.

Figure 31 empirically verifies two consequences of this conjecture. In Figure 31(a), we show that first-layer weights with large alignment with standard basis vectors also have large second-layer weights, indicating that axis-aligned first-layer weights are highly influential in the final model's prediction. Figure 31(b) shows that the biases in first-layer ReLU units are predominantly negative.

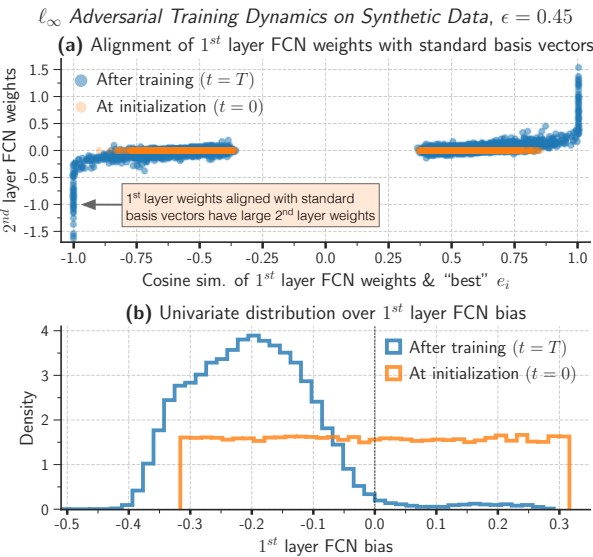

Figure 31: Adversarial training dynamics for training one-hidden-layer FCNs with width $50,000$ on 10-dimensional synthetic data. Subplot (a) shows that first-layer neurons aligned with standard basis vectors have large second-layer weights. Given a normalized $1^{st}$ layer weight vector (i.e., rescaled so that it has unit $\ell_2$ norm), the x-axis in (a) plots the coordinate with largest magnitude in this normalized vector. Note that the gap around origin is due to that fact that the largest magnitude coordinate in a unit $\ell_2$ norm vector in $d = 10$ dimensions is at least $\frac{1}{\sqrt{d}} \approx 0.32$. Subplot (b) shows that $\ell_\infty$ adversarial training results in first-layer bias terms that are predominantly negative. These observations support Conjecture 1 and indicate that adversarial training quickly enters the rich regime.

The following lemma shows that the input gradients of $\tilde{\nu}$ in (9) indeed highlight $j^*(x)$.

**Lemma 1.** *Let data distribution $\mathcal{D}$ follow (2) with $\tilde{d} = 1, \eta = 0$ and $u^* = 1$ and let $\tilde{\nu}$ be as defined in (9). Then, for any $(x, y) \sim \mathcal{D}$, we have: $\nabla_x \mathcal{L}(\tilde{\nu}, (x, y)) = c \cdot e_{j^*(x)}$, where $c \neq 0$ is a constant.*

Assuming Conjecture 1, this lemma shows that for the special case $\tilde{d} = 1$ and $\eta = 0$, adversarially trained models have input gradients that reveal instance-specific features important for classification. Conjecture 1 and Lemma 1 also explain several other empirically observed properties of adversarial training such as visually perceptible input gradients and adversarial examples [29]. In this section, we prove Lemma 1.

*Proof of Lemma 1.* Given the classifier $\tilde{\nu}$ and a data point $(x, y)$, the input gradient is given by

$$\nabla_x \mathcal{L}(\tilde{\nu}, (x, y)) = c \cdot \nabla_x f(\tilde{\nu}, x)$$
$$= c \cdot \mathbb{E}_{(w,r,b) \sim \tilde{\nu}} \left[ w\phi' \left( \langle r, x \rangle + b \right) r \right],$$

where $c = \frac{-y \exp(-y \cdot f(\tilde{\nu}, x))}{1 + \exp(-y \cdot f(\tilde{\nu}, x))}$. Recall from (9) that

$$\tilde{\nu} = \frac{1}{d} \sum_{i \in [d/2]} \delta_{\theta_i} + \delta_{\theta_i'},$$

where $\theta_i := (\frac{1}{\sqrt{2}}, \frac{3}{\sqrt{20}} e_i, \frac{-1}{\sqrt{20}})$ and $\theta_i' := (\frac{-1}{\sqrt{2}}, \frac{-3}{\sqrt{20}} e_i, \frac{-1}{\sqrt{20}})$, $e_i$ denotes the $i^{\text{th}}$ standard basis vector in $\mathbb{R}^d$. If $(x, y) = (y e_{j^*(x)}, y)$ and $(w, r, b) \sim \delta_{\theta_i}$ or $(w, r, b) \sim \delta_{\theta_i'}$, then $\Pr \left[ \phi' \left( \langle r, x \rangle + b \right) \neq 0 \right] > 0$ if and only if $i = j^*(x)$. Consequently, the only contribution the input gradient comes from $\delta_{\theta_{j^*(x)}}$ and $\delta_{\theta_{j^*(x)}'}$. So,

$$\nabla_x \mathcal{L}(\tilde{\nu}, (x, y)) = c \cdot \mathbb{E}_{(w,r,b) \sim \tilde{\nu}} \left[ w\phi' \left( \langle r, x \rangle + b \right) r \right]$$
$$= c' \cdot \left( \mathbb{E}_{(w,r,b) \sim \delta_{\theta_{j^*(x)}}} \left[ w\phi' \left( \langle r, x \rangle + b \right) r \right] + \mathbb{E}_{(w,r,b) \sim \delta_{\theta_{j^*(x)}'}} \left[ w\phi' \left( \langle r, x \rangle + b \right) r \right] \right)$$
$$= c'' \cdot e_{j^*(x)}.$$

This proves the result. $\qquad\square$