# OpenReview forum: "Do Input Gradients Highlight Discriminative Features? "
_NeurIPS.cc/2021/Conference — NeurIPS 2021 Poster_

### Official Review · Reviewer_8z9h · 2021-07-05

**Rating:** 6
**Confidence:** 3

**Summary:**

This paper studies the validity of a key assumption (i.e. Assumption A) made in popular post-hoc attribution methods empirically and theoretically. Comprehensive experiments are conducted with specifically designed BlockMNIST. Moreover, the authors constructs a simplified setting for theoretical analysis to demonstrate feature leakage.

**Limitations And Societal Impact:**

A few limitations can be the restructre of the introduction section: the paper is a bit hard to follow with terms from the interpretability area and the intro part is introducing related works and case studies without a clear storyline.

Moreover, the feature leakage hypothesis is not very well justified in the BlockMNIST setting.

More investigations about whether the probes in this work can be leveraged to shed light on the predictive features containing group-sensitive attributes (male vs female) should be done to improve the potential societal impact of this work.

**Main Review:**

This work provides a series of systematic probes like BlockMNIST into the effects of input gradients in feature attribution. With the extensive studies for evaluating the feature leakage hypothesis for violating the Assumption (A), It provides insights for several aspects like why input gradients of standard models tend to violate assumption (A) including standard and robust models. I'm taken by the way to challenge the common attribution assumptions in interpretability and it will potentially benefit the development of this line of work.

**Time Spent Reviewing:**

6

---

> ### Author Response · Authors · 2021-08-11
> **Response to Reviewer 8z9h (#4)**
>
> Thank you for reviewing our paper! Below, we address your comments regarding presentation and BlockMNIST:
>
> **Writing**
>
> As per your suggestions, we will make two changes in the revised version to improve the readability of the paper. First, we will improve the introduction to provide more context about related work as well as terms from the interpretability literature. Second, we will discuss the potential for negative societal impact due to misleading feature importance estimates of input gradient attributions.
>
> **BlockMNIST and feature leakage**
>
> Could you please elaborate your reasons on why you believe that feature leakage is not well justified in the BlockMNIST setting? As shown in our experiments (Section 5) and theory (Section 6), there is indeed leakage of “features” from one point to another which leads to ineffectiveness of input gradients.

---

> ### Author Response · Authors · 2021-08-26
> **Quick follow-up**
>
> Thank you for taking the time to review our work. We hope that our rebuttal has addressed your questions and concerns. Please let us know if you have any unresolved concerns or additional questions about the paper or our rebuttal!

---

### Official Review · Reviewer_L2GP · 2021-07-15

**Rating:** 6
**Confidence:** 4

**Summary:**

The authors test the current assumption for gradient-based explanation methods, i.e., a high magnitude of vanilla gradients highlight more discriminative task-relevant features and vice-versa. The authors theoretically and empirically test the above assumption using DiffROAR, a new evaluation framework, and BlockMNIST, a new semi-real dataset. Empirical analysis using DiffROAR concludes that gradients of standard models violate the above assumption. whereas gradients of adversarially trained models don't. The BlockMNIST dataset separates the signal and noise part of a given sample by encoding a priori knowledge of discriminative and non-discriminative features in the samples. The work tackles one of the existing problems of gradient-based explanation methods and suggests that the proposed frameworks and datasets can be used as sanity checks to audit instance-specific interpretability methods.

**Limitations And Societal Impact:**

Yes, the authors briefly mention the works' limitations. One of the major drawbacks of this work is its scope. The authors exclusively focus on vanilla input gradients, whereas works like Bansal et al. 2020 have already studied improved gradient variants like SmoothGrad and showed their differences for standard and adversarially trained models. It would be great to show similar findings for other gradient-based explanation methods.

**Main Review:**

Strengths:
1. The paper is well-written with coherent details of the proposed DiffROAR evaluation metric and BlockMNIST dataset.

2. The proposed DiffROAR evaluation metric is an intuitive extension of the previously proposed ROAR metric.

3. The empirical findings that standard one-hidden-layer MLPs violate the gradient assumptions are supported by respective theoretical analyses.

Weaknesses:
1. In the BlockMNIST dataset design, the authors mention -- unlike the MNIST signal block that is fully predictive of the class, the non-discriminative null block contains no information about the class. The argument is not clear because CNNs (both standard and adversarially trained) learn low-level features, such as horizontal and vertical edges, in their earlier layers, and they will be identified even for the null blocks in the dataset. Hence, contributing to the actual predicting class. A sanity check would have been to remove the null blocks from the images and check whether the classification performance of the model remains unchanged.

2. In the feature leakage hypothesis, the authors mention that standard models violate the gradient assumption because the discriminative signal blocks are randomly placed in the images and propose the BlockMNIST-Top dataset variant that fixed the location of the signal block. Some open questions regarding this experiment are: a) was the feature leakage due to the backpropagated gradients taken from a batch of images? One can expect that using a larger batch size will help ameliorate the effect of placing signal blocks in random locations, b) was the two models trained on BlockMNIST and BlockMNIST-Top trained using the same weight initializations?, and c) the feature leakage hypothesis contradicts that CNNs are invariant to translation, i.e., if we translate the inputs the CNN will still be able to detect the class to which the input belongs.

3. Bansal et al. 2020 analyzed the differences between the vanilla gradients of standard and robust models with similar findings that input gradients of adversarially trained models are more stable than their standard counterparts. Chen et al. 2020 hypothesize that robust models learn smoother kernels as feature extractors, and hence the gradients are stable with respect to non-discriminatory signals.

4. Previous works have detailed the disadvantages of replacing masked features with heuristics such as gray color, blurring, or random noise. It would be great if the authors can comment about the importance of choosing the removing operator in their analysis.

[1] Chen et al., The shape and simplicity biases of adversarially robust ImageNet-trained CNNs, 2020.
[2] Bansal et al., SAM: The sensitivity of attribution methods to hyperparameters, 2020.

**Time Spent Reviewing:**

9

---

> ### Author Response · Authors · 2021-08-11
> **Response to Reviewer L2GP (#3)**
>
> Thank you for taking the time to carefully review our work. Below, we first address your questions about BlockMNIST, translation invariance, and removal operation in DiffROAR. Then, we discuss connections to Bansal et al. and Chen et al., and justify our decision to focus on vanilla input gradient attributions.
>
> **Sanity check for null block in BlockMNIST**
>
> As mentioned in line 265, we already performed the sanity check that you suggest and observed that removing / zeroing out the null block from all images does not change model performance at all. In fact, not only the test accuracy and AUC metric remains unchanged at 100%, but even the logits remain almost unchanged (both for MLPs as well as Resnet18), thereby implying that BlockMNIST models do not rely on the null block. Also, intuitively, if the same null pattern appears in both classes then the CNN activations might fire at the lower layers, but at higher layers, they will be completely ignored to ensure small loss on the training set.
>
>
> **BlockMNIST clarifications**
> 1. *“was the feature leakage…batch of images?”*: Larger batch sizes do not mitigate feature leakage in BlockMNIST. To verify this, we re-ran BlockMNIST experiments with full-batch gradient descent and observed that input gradients of standard MLPs and Resnet18 models still exhibit significant feature leakage. We can add these results in the appendix in the paper.
> 2. *“was the two models…same weight initializations?”*: The results in the paper used the same (default PyTorch) weight initialization scheme but not the same randomness. We also reran the experiment using the same randomly initialized weights to train on both BlockMNIST and BlockMNIST-Top and have obtained the same results presented in the paper. We can clarify this in the future version of the paper.
>
> **Translation invariance and BlockMNIST models**
>
> Intuitively, CNNs are translation invariant only if the object of interest is not closer to the boundary than the receptive field of the final layer; in our case the digits are either close to the top boundary or the bottom boundary and the receptive field of Resnets is quite large. Hence, translation invariance would not hold. This is further supported by recent work [P1] published at CVPR 2020 that demonstrates: “CNNs can and will exploit the absolute spatial location by learning filters that respond exclusively to particular absolute locations by exploiting image boundary effects”.
>
> We observe this phenomenon empirically in our BlockMNIST-Top experiments as well. That is, while models trained on BlockMNIST-Top data (i.e., MNIST digit in top block) attain 100% test accuracy on BlockMNIST-Top images, the accuracy of these models degrades to ~55% (5% better than random chance) when evaluated on BlockMNIST-Bottom images, wherein the MNIST digit (signal) is placed in the bottom block.
>
> **Choice of removal operator in DiffROAR**
>
> Recall that to compute DiffROAR (and top/bottom-k predictive power), we retrain new models from scratch on unmasked datasets. Since the same removal operation is applied to unmask every image (across classes), the choice of removal operator has no effect on our DiffROAR results in Section 4.  To verify this, we evaluated DiffROAR on CIFAR-10 with another removal operator in which pixels are masked/replaced by random gaussian noise (instead of zeros) and observed that the results do not change (i.e., same as in Figure 3). We will add these results to the appendix of the revised version of the manuscript.
>
> **Connections to explanation stability (Bansal et al. 2020 and Chen et al. 2020)**
>
> Thank you for sharing both papers, we will certainly cite them in the revised version. While these papers are relevant, stability and correctness (also known as fidelity, which is our focus) are two distinct desirable properties of explanations [P2]. That is, stability does not imply fidelity—an input-agnostic constant explanation is stable but lacks fidelity. Conversely, fidelity does not imply stability—if the underlying model is itself unstable, then any correct high-fidelity explanation of that model must also be unstable.
>
> While Bansal et al. and Chen et al. identify and explain why input gradients of adversarially trained models are more **stable** compared to those of standard models, our work focuses on identifying and explaining why input gradients of adversarially trained models have more **fidelity** compared to those of standard models. Furthermore, we also take the first step towards theoretically showing that adversarial robustness can provably improve input gradient fidelity (see Appendix E).
>
> **Why focus on vanilla input gradients?**
>
> As discussed in the introduction, several feature attributions such as guided backprop and integrated gradients that output visually sharper attribution maps already fail basic sanity checks such as model randomization and label randomization [12, 13, 20, 21]. So, we chose to focus on vanilla input gradients because (i) vanilla input gradients pass both sanity checks mentioned above and (ii) the input gradient operation is the key building block of all gradient-based attribution methods. Our experiments on real and semi-real data were designed specifically to identify and verify feature leakage as the key issue for input gradients of standard models. However, it is possible that other attribution methods fail due to different reasons. For example, backpropagation-modified explanations fail [19] due to reasons that do not affect vanilla input gradients. Therefore, thoroughly evaluating and identifying failure modes of other feature attribution methods is a significant endeavor in itself and hence we decided to defer it to future work. Nevertheless, in the next revision, we will use our experiments to test whether other methods such as SmoothGrad suffer from feature leakage.
>
> We hope that we have addressed your concerns satisfactorily and are happy to engage in further discussion.
>
> ---
>
> [P1] Kayhan, O.S. and Gemert, J.C.V., 2020. On translation invariance in cnns: Convolutional layers can exploit absolute spatial location. In Proceedings of the IEEE/CVF Conference on Computer Vision and Pattern Recognition (pp. 14274-14285).
>
> [P2] Yeh, C.K., Hsieh, C.Y., Suggala, A., Inouye, D.I. and Ravikumar, P.K., 2019. On the (in) fidelity and sensitivity of explanations. Advances in Neural Information Processing Systems, 32, pp.10967-10978.

---

> > ### Comment · Reviewer_L2GP · 2021-08-25
> > **Discussion response**
> >
> > Dear Authors,
> >
> > Thank you for providing a detailed rebuttal response and additional experiments. While I agree with the arguments (empirically shown results) regarding the translation invariance of BlockMNIST models, I am not convinced that removal operators do not play a role in the output results. Past works have clearly shown the importance of removal operator in the evaluation performance of post-hoc interpretability methods, where using random Gaussian noise as removal operator affects the model predictive behavior. All my other concerns have been addressed and I am happy to increase my score. Thanks!

---

> > > ### Author Response · Authors · 2021-08-25
> > > **Post-rebuttal response to Reviewer L2GP (#3)**
> > >
> > > Thank you for providing a positive response after reviewing our rebuttal!
> > >
> > > _"Past works have clearly...model predictive behavior"_: Could you please share the works that study the effect of removal operators on the evaluation of interpretability methods? As discussed in our rebuttal, we computed our evaluation metric DiffROAR (and ROAR) using two removal operators—gaussian noise and zeros—and obtained the same empirical results. Do you have any suggestions for other removal operators that we should try out?
> > >
> > > Thanks again!

---

> > > > ### Comment · Reviewer_L2GP · 2021-08-25
> > > > **References**
> > > >
> > > > The idea was not to add more experiments and yet another removal operator but to understand why don't we observe any change when using Gaussian noise and zeros as removal operators. Concerning previous works, the following are a bunch of papers that discuss the importance of feature removal techniques in XAI:
> > > >
> > > > 1. Agarwal, C., & Nguyen, A. (2020). Explaining image classifiers by removing input features using generative models. In Proceedings of the Asian Conference on Computer Vision.
> > > >
> > > > 2. Singla, S., Pollack, B., Wallace, S., & Batmanghelich, K. (2021). Explaining the black-box smoothly-a counterfactual approach. In arXiv.
> > > >
> > > > 3. Qiu, L., Yang, Y., Cao, C. C., Liu, J., Zheng, Y., Ngai, H. H. T., ... & Chen, L. (2021). Resisting Out-of-Distribution Data Problem in Perturbation of XAI. In arXiv.
> > > >
> > > > 4. Covert, I., Lundberg, S., & Lee, S. I. (2020). Explaining by removing: A unified framework for model explanation. In arXiv.
> > > >
> > > > I will wait for the response from other reviewers before I change my score. Thanks!

---

> > > > > ### Author Response · Authors · 2021-08-26
> > > > > **On feature removal techniques / removal operators in XAI**
> > > > >
> > > > > Thank you for following up and sharing relevant papers on feature removal techniques!
> > > > >
> > > > > In these papers, the predictive behavior of the **model trained on the original dataset** is used to evaluate properties of interpretability methods. As noted in your response, the choice of feature removal operator can certainly alter the final evaluation results _in this setting_.
> > > > >
> > > > > In contrast, our evaluation metric is based on the remove-and-retrain (ROAR) framework in [P1]. In this approach, the predictive power of **a new model retrained on the unmasked dataset** (i.e, data points after removal operation) is used to evaluate the fidelity of interpretability methods. Note that this approach employs retraining to account for and nullify distribution shifts induced by feature removal operators such as gaussian noise, zeros etc.
> > > > >
> > > > > We will cite previous works that you have shared and clarify this aspect (retraining vs. feature removal operator) in the revised version of our paper. Thanks!
> > > > >
> > > > > ---
> > > > >
> > > > > [P1] Hooker, S., Erhan, D., Kindermans, P.J. and Kim, B., 2018. A benchmark for interpretability methods in deep neural networks. NeurIPS 2019.

---

> > > ### Author Response · Authors · 2021-08-31
> > > **Quick follow-up**
> > >
> > > Dear reviewer,
> > >
> > > A gentle ping about reconsidering your score for the paper. Hopefully, our follow-up response re: removal operators clarifies your pending concern.
> > >
> > > Thanks

---

### Official Review · Reviewer_ENet · 2021-07-15

**Rating:** 7
**Confidence:** 5

**Summary:**

This paper investigates whether input gradients (of robust and normally trained models) highlight discriminative features. As part of this investigation, it introduces DiffROAR (a modified version of ROAR for evaluating the accuracy of feature discriminativeness rankings), as well as the BlockMNIST dataset (plus a simplified version where they perform theoretical analysis). It introduces and discusses the idea of feature leakage as an explanation for why non-robust model gradients sometimes highlight non-discriminative features.

**Limitations And Societal Impact:**

I have a few suggestions for improvements, that could increase the impact of the paper:

- The paper seems to take it as a given that input gradients _ought to_ highlight discriminative features (and certainly many have assumed that), but maybe it's worth stepping back from that even further. In the case of BlockMNIST, it makes total sense that a model might learn to independently detect the same pattern in both sides of the image, and be sensitive to changes that would create a digit in the null block (which is what input gradients actually communicate -- that the model is sensitive to counterfactual changes that would introduce a new digit, even if that's in an unexpected place). To me, these hypothetical features seem totally worthy of being called "discriminative"; the only real distinction is that adding them would take us off the data manifold.

- Relatedly, recent research (e.g. <https://arxiv.org/pdf/1906.00945.pdf> and <https://arxiv.org/pdf/2106.10151.pdf>) suggests that adversarial training encourages model gradients and nearby decision boundaries to lie within the data manifold. In the case of BlockMNIST, that seems like it would likely prevent feature leakage (by preventing model gradients from adding phantom second digits to the null block, and instead forcing them to make more realistic changes to the signal block), and it's even more clear in the simplified 10-feature case (anything with multiple +1s or -1s in the signal block is off the data manifold). I think it's worth discussing this idea because it provides a potential explanation for _why_ adversarial robustness prevents feature leakage (which the paper defers to future work, but doesn't have to). This would really help complete the story and improve the usefulness of the work.

-  While I agree with the authors that DiffROAR is a very useful tool for investigating assumption (A) in this particular context, it might be worth clarifying its limitations if others wish to reuse it in other contexts. For example, consider a redundant dataset where features are either all negative (in which case y=0) or all positive (in which case y=1). In such cases, no feature is more or less informative than any other, so no information can be gained by ranking or removing features. (It also might be worth including some results in the appendix comparing DiffROAR to vanilla ROAR, though it's not critical.)

**Main Review:**

**Originality:** Overall, I think the paper meets the originality threshold. Although DiffROAR is only a slight modification of ROAR, the paper mainly uses it as an analytic tool to investigate an important question in a novel way. Additionally, the simplified and full versions of BlockMNIST are novel.

**Quality:** The submission seems technically sound, with extensive experiments and theoretical analysis. I appreciate how they use multiple distinct methods for testing whether features are truly discriminative (synthetic datasets with ground truth and real data with DiffROAR) to reach consistent and coherent conclusions.

**Clarity:** The submission is well-written; every component was easy to understand, and I feel confident that its results are reproducible to readers.

**Significance:** I think the results are reasonably significant, and help advance our current state of understanding on both input gradients and adversarial robustness (in a novel way). However, I do think the work can and should do more to provide a fuller understanding of its results, particular around _why_ robustness prevents feature leakage; I'll elaborate in the next section.

**Time Spent Reviewing:**

2

---

> ### Author Response · Authors · 2021-08-11
> **Response to Reviewer ENet (#2)**
>
> Thank you for taking the time to carefully review our work. Below, we address your comments  about counterfactual changes, input gradients and data manifold, and the pitfalls of ROAR-based frameworks.
>
> **Counterfactual changes in the null block of BlockMNIST images**
>
> We agree that input gradients can indeed communicate counterfactual changes in the null block to which the model is sensitive. In fact, as evidenced in BlockMNIST experiments, this is basically what we term feature leakage. While this phenomenon seems natural in hindsight, it can be misleading in the context of feature attributions. For example, consider the typical use case for feature attributions: to highlight regions within the given instance/image that are most relevant for model prediction. Now, in the BlockMNIST setting, if input gradients leak digit-like features into the null block, then the feature attributions in the null block can be easily (mis)interpreted as the non-discriminative null patch being highly relevant for model prediction.
>
> **Adversarial training and data manifold**
>
> Thank you for the suggestion. Indeed the simplified 10-feature dataset might help make the connection between feature leakage and input gradients of standard models falling off the data manifold more concrete. We will expand upon this in the next revision.
>
> In Appendix E, we conjecture the precise form of adversarially trained models on the simplified 10-feature dataset and then show that this avoids feature leakage. It turns out that the same conjecture can also be used to show that adversarially trained models have input gradients that lie on the data manifold, thereby closely relating avoidance of feature leakage with input gradients lying close to the data manifold. While we justify our conjectured characterization of adversarially trained models using numerical experiments, proving it rigorously seems quite challenging.
>
> **Pitfalls of ROAR (and DiffROAR)**
>
> We agree with your counterexample; ROAR-based frameworks are not useful in settings wherein every feature is equally informative (and learned by the model) because no information can be gained by removing + retraining on a subset of these features. In our opinion, another pitfall of ROAR and DiffROAR is the key assumption that models retrained on unmasked datasets learn the same features as the model trained on the original dataset. In the absence of ground-truth features, this assumption is empirically supported by findings that suggest that different runs of models sharing the same architecture learn similar features [P1, P2]. Nevertheless, this is still an assumption. This limitation is precisely why we decided to verify and better understand our DiffROAR findings via empirical (Section 5) and theoretical analysis (Section 6) in two settings wherein ground-truth discriminative features are known a priori. We will certainly discuss both limitations in more detail in the revised version.
>
> We hope we have satisfactorily addressed your concerns. Happy to provide further clarifications if needed.
>
> ---
>
> [P1] Hacohen, G., Choshen, L. and Weinshall, D., 2020, Let’s agree to agree: Neural networks share classification order on real datasets.
>
> [P2] Li, Y., Yosinski, J., Clune, J., Lipson, H. and Hopcroft, J.E., 2015. Convergent learning: Do different neural networks learn the same representations?

---

> > ### Comment · Reviewer_ENet · 2021-08-11
> > **Response to response**
> >
> > This response addresses all of my outstanding concerns, and I'm excited to see an expanded/more explicit discussion of the relationship between feature leakage and manifold misalignment! Again, I think this is a solid submission that helps improve our (or at least my) understanding of both adversarial robustness and feature attributions.

---

### Official Review · Reviewer_TbxU · 2021-07-16

**Rating:** 6
**Confidence:** 3

**Summary:**

The authors present a procedure to check the relative influence on model prediction between pixels with largest input gradient magnitudes versus pixels with lowest input gradient magnitudes. This is an extension of ROAR, which trains a model on data with certain pixels masked out. The authors evaluate DiffROAR on 4 image datasets and 1 synthetic dataset, showing that more robust models are more likely to have input gradients on relevant pixels than non-robust models. They also present a theoretical analysis showing that a standard MLP with one hidden layer can violate the assumption that higher input gradients mean more relevant pixels.

**Limitations And Societal Impact:**

The limitations were very briefly discussed at the end. It was surprising to me that the authors chose to solely focus on input gradients, when there are so many other existing feature attribution methods, as cited by the authors. They did mention expanding to these as future work.

There is no discussion on potential negative societal impact. One example societal connection: because the authors are pointing out a flaw in the connection between input gradients and true feature importance, input gradients should not be a trusted measure of pixel importance (so if the claim that model is using "fair" features is based on input gradients, that should be taken with skepticism).

**Main Review:**

*** Originality & Significance ***: The authors do a thorough job with citing related works. The very closely-related prior work of ROAR differs in one main aspect: ROAR checks the relevance of pixels with high measured importance, and DiffROAR compares the importance of those with pixels with low measured importance. The goal of the paper seems limited, in that it seems two key claims have already been established in prior work - to quote from the abstract:

"Our results suggest that (i) input gradients of standard models (i.e., trained on original data) may grossly violate (A), whereas (ii) input gradients of adversarially robust models satisfy (A) reasonably well."

As supported in Line 112,  (i) is established in the related ROAR work:

"...the ROAR framework show[s] that multiple attribution methods, including vanilla input gradients, are no better than model-independent random attributions that lack explanatory power"

And as supported in Line 120, (ii) is established in the work of Kim et. al. [cited as 34]:

"Additionally, Kim et al. [34] use the ROAR framework to conjecture that adversarial training “tilts” input gradients to better align with the data manifold."

The authors write that their goal here is to augment ROAR "to understand when and why input gradients violate assumption (A)". This is partly addressed in their section on experiments (Section 4), by showing input gradients are more often higher magnitude on relevant pixels with robust models than in standard models. However, [34] has already addressed this in much more detail (the boundary tilting theory).

On the other hand, the theory in Section 6 seemed more significant to me. I am not as well versed in the theory literature in this area as compared to feature attribution, and cannot speak to originality very well. But, the result was very interesting - it's possible for a max-margin classifier (of a one-hidden-layer MLP) to place equal input gradient over locations where relevant features have appeared in the data, and 0 gradient on locations holding only noise. It was interesting to see this happen in BlockMNIST as well.

I am recommending a score of 5, because the majority of the paper's goal seems to have been already considerably explored in past papers (i.e., addressed the exact question of the paper's title), as evidenced in the related works section. In addition, the authors' stated contribution of exploring "when and why input gradients violate assumption (A)" seems to have been addressed more thoroughly in the paper cited as [34].

*** Clarity ***: The paper is very well-written, with only minor typos (e.g. missing period on Line 363). I thought it was enjoyable to read, and well-thought-out in its organization. The theory section was confusing at first - I didn't understand the difference between task-relevant and instance-specific, yet also task-relevant, features (e.g., the sentence on Lines 306-308 was not clear at all). Yet, Figure 5 cleared it up.

**Time Spent Reviewing:**

4

---

> ### Author Response · Authors · 2021-08-11
> **Response to Reviewer TbxU (#1)**
>
> Thank you for taking the time to carefully review our work. Below, we address your concerns about the novelty of our results vis-a-vis work of Kim et al. [34] and Hooker et al. [14], and justify our decision to focus on vanilla input gradient attributions.
>
> **Novelty vis-a-vis Kim et al. [34]**
>
> First, unlike the empirical analysis with a small CNN model on CIFAR-10 in [34], we thoroughly establish our DiffROAR results across datasets/architectures/hyperparameters, revealing a significantly larger gap between attribution quality of standard vs. robust models compared to that reported in [34]. More importantly, it seems that the analysis in [34] is based on a disproved conjecture; we provide details below:
> - *Boundary tilting conjecture in Kim et al.*: Motivated by the boundary tilting hypothesis by Tanay and Griffin [P5], Kim et al.use a two-dimensional synthetic dataset to empirically show that the decision boundary of robust models aligns better with the vector between the two class-conditional means. Unfortunately, this empirical evidence might be misleading, as [P1] theoretically demonstrates that “this exact statement is not true beyond two dimensions” (page 15).
> - *Alternative hypotheses for adversarial examples*: Furthermore, several recent works have also provided concrete evidence to support alternative hypotheses [P1, P2, P3, P4] for the existence of adversarial examples that counter the boundary tilting hypothesis that Kim et al. build upon.
>
> This discrepancy in these results motivates the need for a multipronged (empirical + theoretical) approach, which we adopt to empirically identify the feature leakage hypothesis using BlockMNIST and theoretically verify the hypothesis in Section 6.  To the best of our knowledge, we also take the first step towards theoretically showing that adversarial robustness can provably improve input gradient quality (in Appendix E).
>
>
>
>
> **Novelty vis-a-vis ROAR / Hooker et al. [14]**
>
> The questions below illustrate key differences between ROAR / Hooker et al. [14] and our paper:
> 1. *Does the paper verify assumption (A)?* In Hooker et al, the ROAR framework essentially computes the top-k predictive power only, which is not sufficient to test assumption (A). In our paper,  DiffROAR explicitly compares the top-k and bottom-k predictive power to test whether the given attribution method satisfies assumption (A).
> 2. *Are the results in the paper conclusive?* Both, ROAR and DiffROAR, make a key assumption: models retrained on unmasked datasets learn the same features as the model trained on the original dataset. Although empirically supported [P6, P7], this assumption makes it difficult to conclusively test assumption (A). Therefore, unlike Hooker et al., we empirically (Section 5) and theoretically (Section 6) verify our DiffROAR findings in settings wherein ground-truth features are known a priori.
> 3. *Does the paper identify why standard input gradients violate (A)?* Hooker et al. do not discuss why input gradients lack explanation fidelity. In our paper, we hypothesize *feature leakage* as the key reason for ineffectiveness of input  gradients, and validate it with empirical as well as theoretical analysis on BlockMNIST-based data.
>
> **Why focus on vanilla input gradients?**
>
> As discussed in the introduction, several feature attributions such as guided backprop and integrated gradients that output visually sharper saliency maps already fail basic sanity checks such as model randomization and label randomization [12, 13, 20, 21]. So, we chose to focus on vanilla input gradients because (i) vanilla input gradients pass both sanity checks mentioned above and (ii) the input gradient operation is the key building block of all gradient-based attribution methods. Our experiments on real and semi-real data were designed specifically to identify and verify feature leakage as the key issue for input gradients of standard models. However, it is possible that other attribution methods fail due to different reasons. For example, backpropagation-modified explanations fail [19] due to reasons that do not affect vanilla input gradients. Therefore, thoroughly evaluating and identifying failure modes of other feature attribution methods is a significant endeavor in itself and hence we decided to defer it to future work.
>
> We hope we have satisfactorily addressed your concerns. Happy to provide further clarifications if needed.
>
> ---
>
> [P1] Ilyas, A., Santurkar, S., Tsipras, D., Engstrom, L., Tran, B., & Madry, A. (2019). Adversarial examples are not bugs, they are features.
>
> [P2] Shafahi, A., Huang, W.R., Studer, C., Feizi, S. and Goldstein, T., 2018. Are adversarial examples inevitable?
>
> [P3] Bubeck, S., Lee, Y.T., Price, E. and Razenshteyn, I., 2019, May. Adversarial examples from computational constraints.
>
> [P4] Shah, H., Tamuly, K., Raghunathan, A., Jain, P. and Netrapalli, P., 2020. The pitfalls of simplicity bias in neural networks.
>
> [P5] Tanay, T. and Griffin, L., 2016. A boundary tilting perspective on the phenomenon of adversarial examples.
>
> [P6] Hacohen, G., Choshen, L. and Weinshall, D., 2020, Let’s agree to agree: Neural networks share classification order on real datasets.
>
> [P7] Li, Y., Yosinski, J., Clune, J., Lipson, H. and Hopcroft, J.E., 2015. Convergent learning: Do different neural networks learn the same representations?

---

> > ### Comment · Reviewer_TbxU · 2021-09-02
> > **Response to rebuttal**
> >
> > Hi authors, thank you for addressing the concerns I brought up. It is good that your work includes evaluation across a much larger variety of models & datasets than Kim et. al. [34]. I also was not aware of the prior works you mentioned that show a broader discussion around this topic, which gives evidence for the need to theoretically verify hypotheses, as you stated as well.
> >
> > I hope to see a subset of your experiments evaluated with other input gradient methods such as SmoothGrad as you mentioned to another reviewer. Although input gradients are a "key building block" of some gradient-based attribution methods, they may not be so "key" if the differences that other attribution methods have (in relation to input gradients) have a larger influence in the outcome of the experiments.
> >
> > I will increase my score, as you have highlighted the importance of the theoretical contributions and more extensive experiments.

---

> > > ### Author Response · Authors · 2021-09-03
> > > **Response to Reviewer TbxU**
> > >
> > > Dear reviewer,
> > >
> > > Thank you for the positive response and for increasing your score! Glad to know that our rebuttal addressed most of your questions and concerns.  We will certainly include experiments on SmoothGrad and Integrated Gradients in the updated version of our paper.
> > >
> > > Thanks again!

---

> ### Author Response · Authors · 2021-08-26
> **Quick follow-up**
>
> Thank you for taking the time to review our work. We hope that our rebuttal has addressed your questions and concerns. Please let us know if you have any unresolved concerns or additional questions about the paper or our rebuttal!

---

### Decision · Program_Chairs · 2021-09-27

**Decision:**

Accept (Poster)

**Comment:**

Thank you for your submission to NeurIPS.  The reviewers and I are in agreement that the paper presents interesting new insights into the problem of understanding the significance of input gradients as they concern robust and nonrobust models.  The addition of the BlockMNIST data set seems particularly helpful to better understanding and furthering research in the area.  I'm happy to recommend that the paper be accepted.

Since there was a fair amount of response to reviewer feedback, I'd mainly highlight that many of the comments seemed to center around further (discussion-based, not quantitative) comparison to previous works (such as [34] and Ilyas et a, 2019 mentioned by the reviewer).  Overall I think that relating the results here more to potentially qualify phenomena observed in previous studies, at least to the extent possible without substantial revisions, would improve most upon the current paper.